# PREFERENCE OPTIMIZATION FOR COMBINATORIAL OPTIMIZATION PROBLEMS

## ABSTRACT

Reinforcement Learning (RL) has emerged as a powerful tool for neural combinatorial optimization, enabling models to learn heuristics that solve complex problems without requiring optimal solutions. Despite significant progress, existing RL approaches face challenges such as diminishing reward signals and inefficient exploration in vast combinatorial action spaces, leading to inefficient learning. In this paper, we propose **Preference Optimization (PO)**, a novel framework that transforms quantitative reward signals into qualitative preference signals via statistical comparison modeling, emphasizing the superiority among generated solutions. Methodologically, by reparameterizing the reward function in terms of policy probabilities and utilizing preference models like Bradley-Terry and Thurstone, we formulate an entropy-regularized optimization objective that aligns the policy directly with preferences while avoiding intractable computations. Furthermore, we integrate heuristic local search techniques into the fine-tuning process to generate high-quality preference pairs, helping the policy escape local optima. Empirical results on standard combinatorial optimization benchmarks, such as the Traveling Salesman Problem (TSP), the Capacitated Vehicle Routing Problem (CVRP) and the Flexible Flow Shop Problem (FFSP), demonstrate that our method outperforms traditional RL algorithms, achieving superior sample efficiency and solution quality. Our work offers a simple yet efficient algorithmic advancement in neural combinatorial optimization.

## 1 INTRODUCTION

Combinatorial Optimization Problems (COPs) are fundamental in numerous practical applications, including route planning, circuit design, scheduling, and bioinformatics Papadimitriou & Steiglitz (1998); Cook et al. (1994); Korte et al. (2011). These problems require finding an optimal solution from a finite but exponentially large set of possibilities and have been extensively studied in the operations research community. While computing the exact solution is impeded by their NP-hard complexity Garey & Johnson (1979), efficiently obtaining near-optimal solutions is essential from a practical standpoint.

Deep learning, encompassing supervised learning and reinforcement learning (RL), has shown great potential in tackling COPs by learning heuristics directly from data Bengio et al. (2021); Vinyals et al. (2015). However, supervised learning approaches heavily rely on high-quality solutions, and due to the NP-hardness of COPs, such training data may not guarantee optimality, which can lead models to fit suboptimal policies. In contrast, RL has emerged as a promising alternative, achieving success in areas involving COPs such as mathematical reasoning Silver et al. (2018), chip design Mirhoseini et al. (2021), and discovering efficient algorithms Fawzi et al. (2022). RL leverages neural networks to approximate policies and interactively obtains rewards from environment, allowing models to improve without requiring high-quality solutions Bello et al. (2016); Kool et al. (2019).

Despite its potential, applying RL to COPs presents significant challenges. **Diminishing reward signals**: Current methods often frame the training process at the trajectory level due to the interdependence of actions at different timesteps. As the policy improves, the differences in trajectory-level reward signals between solutions diminish, leading to negligible gradients and slow convergence during later training phases. **Unconstrained action spaces**: The vast combinatorial action spaces complicate efficient exploration, rendering traditional exploration techniques like entropy regular-

ization of trajectory computationally infeasible. **Additional inference time**: While neural solvers are efficient in inference, they often suffer from finding the near-optimal solutions. Many works adopt techniques like local search as a post-processing step to further improve solutions, but this incurs additional computational time during inference.

To address the issue of diminishing reward signals and inefficient exploration, we propose transforming quantitative reward signals into qualitative preference signals, focusing on the superiority among generated solutions rather than their absolute reward values. This approach stabilizes the learning process and theoretically emphasizes optimality, as preference signals are insensitive to the scale of rewards. By deriving our method from an entropy-regularized objective, we inherently promote efficient exploration within the vast combinatorial action spaces of COPs, overcoming the computational intractability associated with traditional entropy regularization techniques. Additionally, to mitigate the extra inference time induced by local search, we integrate such techniques into the fine-tuning process rather than using them as post-processing steps, which enables the policy to learn from improved solutions without incurring additional inference time.

Furthermore, preference-based optimization has recently gained prominence through its application in Reinforcement Learning from Human Feedback (RLHF) for large language models Christiano et al. (2017); Rafailov et al. (2024); Meng et al. (2024). Inspired by these advancements, we introduce a novel update scheme that bridges preference optimization with COPs, leading to a more effective and consistent learning process. In this work, we propose a novel algorithm named Preference Optimization (PO), which can seamlessly substitute conventional policy gradient methods in many contexts. In summary, our contributions are:

1. **A Novel Preference-Based Framework:** We introduce a new framework that transforms quantitative reward signals into qualitative preference signals, stabilizing the learning process and theoretically emphasizing optimality independently of reward scaling.

2. **An Efficient Optimization Objective:** By reparameterizing the reward function in terms of the policy and utilizing preference models such as Bradley-Terry and Thurstone, we formulate an entropy-regularized optimization objective that aligns the policy directly with preferences, avoiding intractable computations of traversing whole action space.

3. **Integration with Appealing Solutions:** We demonstrate the compatibility of our approach with heuristic local search methods by incorporating them into the fine-tuning process. This integration generates high-quality preference pairs and helps the policy escape local optima without incurring additional inference time.

## 2 RELATED WORK

**RL-based Neural Solvers.** The pioneering application of Reinforcement Learning for Combinatorial Optimization problems (RL4CO) by Bello et al. (2016); Nazari et al. (2018); Kool et al. (2019) has prompted subsequent researchers to explore various frameworks and paradigms. We classify the majority of RL4CO research from the following perspectives:

*End-to-End Neural Solvers.* Several works have focused on designing end-to-end neural solvers that directly map problem instances to solutions. Techniques exploiting the inherent equivalence and invariance properties of COPs have been proposed to ease the difficulty in approaching near-optimal solutions Kwon et al. (2020); Kim et al. (2022); Ouyang et al. (2021); Kim et al. (2023). For example, POMO Kwon et al. (2020) utilizes multiple diverse starting points to improve training efficiency, while Sym-NCO Kim et al. (2022) leverages problem symmetries to enhance performance. Other studies have incorporated entropy regularization at the step level to foster exploratory behaviors, thereby improving solution diversity and quality Xin et al. (2021a); Sultana et al. (2020). Additionally, efforts have been made to diversify the training dataset to develop more generalized solvers capable of handling a wider range of problem instances Bi et al. (2022); Wang et al. (2024); Zhang et al. (2022); Zhou et al. (2023); Jiang et al. (2024). While most of these works aim to boost performance through architectural innovations or learning paradigms, less attention has been given to algorithmic advancements in the optimization objectives themselves. For instance, Jin et al. (2023) propose a normalized reward for updating the policy, but this approach still struggles to effectively emphasize optimality in the solutions.

*Hybrid Solvers.* Blending neural methodologies with conventional optimization techniques presents a promising research direction. Such integration incorporates established heuristics like $k$-opt, Ant Colony Optimization, Monte Carlo Tree Search, and the Lin-Kernighan algorithm, enhancing solution quality as demonstrated in d O Costa et al. (2020); Wu et al. (2021); Ye et al. (2023); Xin et al. (2021b). For example, NeuRewriter d O Costa et al. (2020) combines neural networks with local search heuristics for graph rewriting, while NeuroLKH Xin et al. (2021b) integrates deep learning with the LKH algorithm. While such techniques often serve as post-processing steps to refine near-optimal solutions, as in Fu et al. (2021); Ma et al. (2021); Ouyang et al. (2021), the additional inference time interferes with efficiency and may not be suitable for time-critical scenarios.

**Preference-based Reinforcement Learning.** Preference-based reinforcement learning (PbRL) is another area related to our work, which has been widely studied in offline RL settings. PbRL involves approximate the ground truth reward function from preference information rather than relying on explicit reward signals Wirth et al. (2017). This approach is particularly useful when reward signals are sparse or difficult to specify. Recently, works such as Hejna & Sadigh (2024); Rafailov et al. (2024); Meng et al. (2024) have proposed novel paradigms to directly improve the KL-regularized policy without the need for learning an approximate reward function, leading to more stable and efficient training. This has led to the development of a series of works Azar et al. (2024); Park et al. (2024); Hong et al. (2024) in the RLHF phase within language-based models, where preference information is leveraged to fine-tune large language models effectively.

Our work bridges the gap between these domains by introducing a preference-based optimization framework specifically tailored for COPs. By transforming quantitative reward signals into qualitative preferences, we address key challenges in RL4CO, such as diminishing reward differences and exploration inefficiency, while avoiding the need for explicit reward function approximation as in traditional PbRL.

## 3 METHODOLOGY

In this section, we first recap Reinforcement Learning, focusing particularly on for Combinatorial Optimization problems (RL4CO), and Preference-based Reinforcement Learning (PbRL). Next, we explain how to leverage these techniques to develop a novel optimization objective to train efficient neural solvers that rely solely on relative superiority among generated solutions. Subsequently, we investigate the compatibility of our approach with Local Search techniques for solver training. Our work results in a simple and consistent algorithm.

### 3.1 REINFORCEMENT LEARNING FOR COMBINATORIAL OPTIMIZATION PROBLEMS

RL trains an agent to maximize cumulative rewards by interacting with an environment and receiving reward signals. In COPs, the state transitions are typically modeled as deterministic. A commonly used policy gradient method is REINFORCE Sutton & Barto (2018), whose update rule is given by:

$$\nabla_\theta J(\theta) = \mathbb{E}_{x \sim \mathcal{D}, \tau \sim \pi_\theta(\tau | x)} \left[ (r(x, \tau) - b(x)) \nabla_\theta \log \pi_\theta(\tau \mid x) \right]$$
$$\approx \frac{1}{|\mathcal{D}|} \sum_{x \in \mathcal{D}} \frac{1}{|S_x|} \sum_{\tau \in S_x} \left[ (r(x, \tau) - b(x)) \nabla_\theta \log \pi_\theta(\tau \mid x) \right], \quad (1)$$

where $\mathcal{D}$ is the dataset of problem instances, $x \in \mathcal{D}$ represents an instance, $S_x$ is the set of sampled solutions (trajectories) for $x$, $r(x, \tau)$ is the reward function derived from distinct COPs and $b(x)$ represents the baseline used to calculate the advantage function $A(x, \tau) = r(x, \tau) - b(x)$, which helps reduce the variance of the gradient estimator. The policy $\pi_\theta(\tau \mid x)$ defines a distribution over trajectories $\tau = (a_0, a_1, \ldots, a_T)$ given the instance $x$. Each trajectory $\tau$ is a sequence of actions generated by the policy: $\pi_\theta(\tau \mid x) = \prod_{t=0}^{T} \pi_\theta(a_t \mid s_t)$, with $s_0$ being the initial state determined by $x$, and $s_t$ representing the state at time step $t$, which is a function of previous states and actions (e.g., $s_t = f(s_{t-1}, a_{t-1})$). The action $a_t$ is selected by the policy based on the current state $s_t$.

Unlike popular RL environments such as Atari Bellemare et al. (2013) and Mujoco Todorov et al. (2012), where rewards can vary widely and provide strong learning signals, COPs present unique challenges. As the policy improves, the differences in reward signals between solutions diminish.

Specifically, the agent often obtains solutions with minimal differences in rewards, i.e., $|r(x, \tau) - b(x)| < \epsilon$, where $\epsilon$ is small. This leads to negligible updates to the policy objective $J(\theta)$, which heavily relies on the advantage function $A(x, \tau) = r(x, \tau) - b(x)$. Consequently, the policy struggles to escape local optimum during later training stages.

Furthermore, models in COPs focus on optimizing the expected maximum reward during inference:

$$\underbrace{\mathbb{E}_{x \sim \mathcal{D}} \left[ \max_{\tau \sim \pi_\theta(\tau|x)} r(x, \tau) \right]}_{\text{Inference objective}} \neq \underbrace{\mathbb{E}_{x \sim \mathcal{D}} \left[ \mathbb{E}_{\tau \sim \pi_\theta(\tau|x)} r(x, \tau) \right]}_{\text{Training objective}}.$$

Inconsistency between training objectives (that optimize expected rewards) and inference objectives (which seek the best possible solution, i.e., maximized rewards) can lead to performance degradation. During training, the improvement of the model leads to a gradual reduction in the numerical values of advantage $A(x, \tau)$ in Eq. 1, which weakens the learning signal under the traditional RL framework. Consequently, REINFORCE fails to effectively emphasize optimality. Therefore, it is necessary to construct a more stable reward signal that highlights optimality.

## 3.2 PREFERENCE-BASED REINFORCEMENT LEARNING

In PbRL Wirth et al. (2017), the agent optimizes a learned reward function based on an offline dataset of preferences, rather than directly receiving reward signals through interaction with the environment. We assume access to a preference dataset $\mathcal{D}_p = \{(\tau_1, \tau_2, y)\}$, where each triplet consists of two trajectories $\tau_1$ and $\tau_2$, and a preference label $y \in \{0, 1\}$. Here, $y = 1$ if $\tau_1$ is preferred over $\tau_2$ (i.e. $\tau_1 \succ \tau_2$), and $y = 0$ otherwise.

Preferences are considered to be generated by an underlying (latent) reward function $\hat{r}(x, \tau)$. Various models can be used to relate reward differences to preferences, such as the Bradley-Terry (BT) model, the Thurstone model David (1963), and the Plackett-Luce (PL) model Plackett (1975). These models bridge the gap between the reward function and observed preferences, allowing us to derive an optimization objective to learn the reward function.

In paired preference models like BT and Thurstone, a function $f(\cdot)$ is used to map the difference between rewards into preference probabilities. The preference probability distribution will be:

$$p^*(\tau_1 \succ \tau_2) = f\left(\hat{r}(x, \tau_1) - \hat{r}(x, \tau_2)\right), \tag{2}$$

where BT model adopt the sigmoid function (i.e., $\sigma(x) = (1 + e^{-x})^{-1}$) and Thurstone model adopt the cumulative distribution function $\Phi(x)$ of the standard normal distribution as $f(\cdot)$.

By establishing this relationship, learning the reward function $\hat{r}_\phi(x, \tau)$ can be formulated as a binary classification problem. The objective is to maximize the likelihood of the observed preferences:

$$\min_{\phi} \quad -\mathbb{E}_{(\tau_1, \tau_2, y) \sim \mathcal{D}_p} \left[ y \log p_\phi(\tau_1 \succ \tau_2) \right].$$

Furthermore, by utilizing the learned reward function $r_\phi$, the policy $\pi_\theta$ learned through the existing RL method is expected to satisfy: $\tau_1 \succ \tau_2 \implies \pi_\theta(\tau_1) > \pi_\theta(\tau_2)$, meaning that if trajectory $\tau_1$ is preferred over trajectory $\tau_2$, then the policy assigns a higher probability to $\tau_1$ than to $\tau_2$. This relationship arises because the policy is optimized to maximize expected rewards according to the learned reward function $r_\phi$.

A major challenge faced by PbRL is the collection of reliable preference data. Preference labels $y$ often need to be assessed based on expert knowledge, which can lead to situations of *preference conflicts*. For instance, one might observe cyclic preferences such as $\tau_1 \succ \tau_2$, $\tau_2 \succ \tau_3$, and $\tau_3 \succ \tau_1$, violating transitivity, thus, constructing consistent and transitive preference labels is a critical issue.

## 3.3 PREFERENCE OPTIMIZATION FOR COMBINATORIAL OPTIMIZATION PROBLEMS

The key insight of our method is to transform the quantitative reward signals into qualitative preferences. This transformation stabilize learning process by avoiding the dependency on numerical reward signals and consistently emphasizes optimality.

A challenge in applying RL to COPs is the exponential growth of the state and action spaces with problem size, making efficient exploration difficult. A common approach to encourage exploration is to include an entropy regularization term $\mathcal{H}(\pi_\theta)$ to balance exploitation and exploration:

$$\max_{\pi_\theta} \quad \mathbb{E}_{x \sim \mathcal{D}, \tau \sim \pi_\theta(\tau|x)} \left[ r(x, \tau) \right] + \alpha \mathcal{H} \left( \pi_\theta(\tau \mid x) \right), \tag{3}$$

where $\alpha > 0$ controls the strength of the entropy regularization, and $\mathcal{H}(\pi_\theta(\cdot \mid x)) = -\sum_\tau \pi_\theta(\tau \mid x) \log \pi_\theta(\tau \mid x)$ is the entropy of the policy for instance $x$. However, computing the entropy term $\mathcal{H}(\pi_\theta)$ is intractable in practice due to the exponential number of possible trajectories.

Following prior works Ziebart et al. (2008); Haarnoja et al. (2017), it is straightforward to show that the optimal policy to the maximum entropy-based objective in Eq. 3 has an analytical form:

$$\pi(\tau \mid x) = \frac{1}{Z(x)} \exp \left( \alpha^{-1} r(x, \tau) \right), \tag{4}$$

where the partition function $Z(x) = \sum_\tau \exp \left( \alpha^{-1} r(x, \tau) \right)$ normalizes the policy over all possible trajectories $\tau$. The detailed derivation is included in the Appendix D.1. Although the solution space of COPs is finite and the reward function $r(x, \tau)$ is accessible, computing the partition function $Z(x)$ is still intractable due to the exponential number of possible trajectories. This intractability makes it impractical to utilize the analytical optimal policy directly in practice.

The specific formulation of Eq. 4 implies that the latent reward function $\hat{r}(x, \tau)$ can be reparameterized in relation to the corresponding policy $\pi(\tau \mid x)$, analogous to the approach adopted in Rafailov et al. (2024) for a KL-regularized objective and in Hejna & Sadigh (2024) within the inverse RL framework. Eq. 4 can thereby be rearranged to express the reward function in terms of its corresponding optimal policy $\pi$ for the entropy-regularized objective:

$$\hat{r}(x, \tau) = \alpha \log \pi(\tau \mid x) + \alpha \log Z(x). \tag{5}$$

From Eq. 5, the ground-truth reward function $r$ can be explicit expressed by the optimal policy $\pi^*$ of Eq. 3. Then we can relate preferences between trajectories directly to the policy probabilities. Specifically, the preference between two trajectories $\tau_1$ and $\tau_2$ can be modeled by projecting the difference in their rewards into a paired preference distribution. Note that this analytic expression naturally avoids intractable term $Z(x)$, since $Z(x)$ is a constant w.r.t. the trajectory $\tau$ and cancels out when considering reward differences.

Using the BT or Thurstone models, by substituting Eq. 5 into Eq. 2, the preference probability between two trajectories becomes:

$$p^*(\tau_1 \succ \tau_2 \mid x) = f \left( \alpha \left[ \log \pi(\tau_1 \mid x) - \log \pi(\tau_2 \mid x) \right] \right), \tag{6}$$

By leveraging this relationship, we transform the quantitative reward signals into qualitative preferences in terms of policy $\pi$.

**Proposition 1** *Let $\hat{r}(x, \tau)$ be a reward function consistent with the Bradley-Terry, Thurstone, or Plackett-Luce models. For a given reward function $\hat{r}'(x, \tau)$, if $\hat{r}(x, \tau) - \hat{r}'(x, \tau) = h(x)$ for some function $h(x)$, it holds that both $\hat{r}(x, \tau)$ and $\hat{r}'(x, \tau)$ induce the same optimal policy in the context of an entropy-regularized reinforcement learning problem.*

Based on Proposition 1, we can conclude that shifting the reward function by any function of the instance $x$ does not affect the optimal policy. This ensures that canceling out $Z(x)$ still preserves the optimality of the policy learned, we defer the proof to Appendix D.2.

We adopt the ground truth reward function $r$ to generate conflit-free preference labels $y = \mathbf{1}_{[\cdot]} : \mathbb{R} \to \{0, 1\}$. As the reward function $r(x, \tau)$ in COPs can be seen as a physical measure, pairwise comparisons generated in this manner preserve a consistent and transitive partial order of preferences throughout the dataset. Moreover, while traditional RL methods may rely on affine transformations to scale the reward signal, our approach benefits from the affine invariance of the preference labels. Specifically, the indicator function is invariant under positive affine transformations:

$$\mathbf{1}_{[k \cdot r(x, \tau_1) + b > k \cdot r(x, \tau_2) + b]} = \mathbf{1}_{[r(x, \tau_1) > r(x, \tau_2)]},$$

for any $k > 0$ and any real number $b$. This property implies that our method emphasizes optimality independently of the scale and shift of the explicit reward function, facilitating the learning process by focusing on the relative superiority among solutions rather than their absolute reward values.

To make the approach practical, we approximate the optimal policy $\pi^*$ with a parameterized policy $\pi_\theta$. This approximation allows us to reparameterize the latent reward differences using $\pi_\theta$, naturally transforming the policy optimization into a classification problem analogous to the reward function trained in PbRL. Guided by the preference information from the ground truth reward function $r(x, \tau)$, the policy optimization objective can be formulated as:

$$\max_\theta \quad J(\theta) = \mathbb{E}_{x \sim \mathcal{D}, (\tau_1, \tau_2) \sim \pi_\theta(\cdot | x)} \left[ \mathbf{1}_{[(r(x, \tau_1) > r(x, \tau_2))]} \cdot \log p_\theta(\tau_1 \succ \tau_2 \mid x) \right], \tag{7}$$

while instantiating with BT model $\sigma(\cdot)$, maximizing $p(\tau_1 \succ \tau_2 \mid x) = \sigma(\hat{r}_\theta(x, \tau_1) - \hat{r}_\theta(x, \tau_2))$ leads to the gradient:

$$\nabla_\theta J(\theta) \approx \frac{\alpha}{|\mathcal{D}||S_x|^2} \sum_{x \in \mathcal{D}} \sum_{\tau \in S_x} \sum_{\tau' \in S_x} \left[ (g_{\text{BT}}(\tau, \tau', x) - g_{\text{BT}}(\tau', \tau, x)) \nabla_\theta \log \pi_\theta(\tau \mid x) \right]$$
$$g_{\text{BT}}(\tau, \tau', x) = \mathbf{1}_{[r(x, \tau) > r(x, \tau')]} \cdot \sigma(\hat{r}_\theta(x, \tau') - \hat{r}_\theta(x, \tau)), \tag{8}$$

where $\hat{r}_\theta(x, \tau)) = \alpha \log \pi_\theta(\tau \mid x) + \alpha \log Z(x)$. Taking a deeper look at the gradient level, compared to the REINFORCE algorithm in Eq. 1, the term about $g(\tau, \tau', x) - g(\tau', \tau, x)$ serves as a substitute for the advantage signal. A key finding is that this reparameterized reward signal ensures that if $r(x, \tau_1) > r(x, \tau_2)$, then the gradient will favor increasing $\pi_\theta(\tau_1)$ over $\pi_\theta(\tau_2)$.

---

**Algorithm 1** Preference Optimization for COPs under Bradley-Terry Model

---

1: **procedure** TRAINING(training set $\mathcal{D}$, number of training steps $T$, number of finetune steps $T_{\text{FT}} >= 0$, batch size $B$, reward model $r$, number of local search iteration $I_{\text{LS}}$)
2:     initialize policy network parameter $\theta$ for $\pi_\theta$
3:     **for** $step = 1, \ldots, T + T_{\text{FT}}$ **do**
4:         $x_i \leftarrow$ SAMPLEINPUT($\mathcal{D}$)   $\forall i \in \{1, \ldots, B\}$
5:         $\tau_i = \{\tau_i^1, \tau_i^2, \ldots, \tau_i^N\} \leftarrow$ SAMPLINGSOLUTIONS($\pi_\theta(x_i)$)   $\forall i \in \{1, \ldots, B\}$
6:         *// Combined with local search for fine-tuning* **(Optional)**
7:         **if** $step > T$ **then**
8:             $\{\hat{\tau}_i^1, \hat{\tau}_i^2, \ldots, \hat{\tau}_i^N\} \leftarrow$ LOCALSEARCH($\tau_i, r, I_{\text{LS}}$)   $\forall i \in \{1, \ldots, B\}$
9:             $\tau_i \leftarrow \tau_i \cup \{\hat{\tau}_i^1, \hat{\tau}_i^2, \ldots, \hat{\tau}_i^N\}$
10:        **end if**
11:        *//Calculate conflict-free preference labels via ground truth reward function $r(x, \tau)$*
12:        $y_{j,k}^i \leftarrow$ PAIRWISEPREFERENCELABEL($\mathbf{1}_{\left[ r(x_i, \tau_i^j) > r(x_i, \tau_i^k) \right]}$)   $\forall j, k \in \{1, \ldots, |\tau_i|\}$
13:        *//Approximating the gradient according to Eq. 8*
14:        $\nabla_\theta J(\theta) \leftarrow \frac{\alpha}{B|\tau_i|^2} \sum_{i=1}^B \sum_{j=1, k=1}^{|\tau_i|} \left( g(\tau_i^j, \tau_i^k, x_i) - g(\tau_i^k, \tau_i^j, x_i) \right) \nabla_\theta \log \pi_\theta(\tau_i^j \mid x_i)$
15:        $\theta \leftarrow \theta + \nabla_\theta J(\theta)$
16:     **end for**
17: **end procedure**

---

### 3.4 COMPATIBILITY WITH LOCAL SEARCH (OPTIONAL)

To further enhance the quality of generated solutions, we investigate the compatibility of PO with heuristic Local Search (LS) techniques, which are widely used to iteratively improve existing solutions generated by traditional or neural solvers. Local search methods have the property of monotonic improvement for fine-tuning existing solutions, which means that for any $\tau$, the improved solution $\text{LS}(\tau)$ satisfies $r(x, \text{LS}(\tau)) \geq r(x, \tau)$ through small adjustments to $\tau$.

Typically, during evaluation, LS is applied as a post-processing step, which can introduce additional inference time due to the multiple iterations required for convergence. To maintain time efficiency during inference while still benefiting from the improvements provided by LS, we propose integrating LS into the solvers' training process rather than serving it as post-processing techniques.

Our proposed Preference Optimization (PO) algorithm relies on the comparison of superiority between trajectories $\tau$. By incorporating LS into fine-tuning, high-quality preference pairs close to

optimality can be generated. Specifically, for each solution $\tau$ generated by the neural solver, we apply a small number of LS iterations to obtain an improved solution $\text{LS}(\tau)$. In most cases, $\text{LS}(\tau)$ is preferred over $\tau$, i.e., $r(x, \text{LS}(\tau)) > r(x, \tau)$, except when LS fails to find an improved solution.

We then form preference pairs $(\tau, \text{LS}(\tau), y)$, where $y = \mathbf{1}_{[r(x, \text{LS}(\tau)) > r(x, \tau)]}$. Our policy optimization objective becomes:

$$\max_{\theta} \quad J(\theta) = \mathbb{E}_{x \sim \mathcal{D}, \tau \sim \pi_\theta(\cdot | x)} \left[ y \cdot \log p_\theta(\text{LS}(\tau) \succ \tau \mid x) \right], \tag{9}$$

where $p_\theta(\text{LS}(\tau) \succ \tau \mid x) = f\left(\alpha \left[\log \pi_\theta(\text{LS}(\tau) \mid x) - \log \pi_\theta(\tau \mid x)\right]\right)$, similar to Eq. 6.

By incorporating these preference pairs into the policy optimization, higher probabilities are encouraged to assign to solutions that are improved by LS. This serves the purpose that incorporating LS during training helps the neural solver escape from local optima, especially during later stages when gradient updates may become less effective due to diminishing differences in reward signals.

It is worth noting that integrating LS introduces additional computational overhead due to the extra LS iterations applied to each sampled trajectory. However, by controlling the number of LS iterations and limiting them to a small number, the additional computational cost can be managed. This trade-off is justified by the significant benefits in learning efficiency and solution quality obtained through this integration. The algorithm is summarized in Algorithm 1.

Combining LS with the proposed PO method, we leverage the strengths of both neural solvers and local search techniques. The neural solver benefits from the fine-tuning capabilities of LS, while maintaining time efficiency during inference by **avoiding** the need for LS as a post-processing step. This synergy leads to more effective learning and improved final solutions.

## 4 EXPERIMENTS

In this section, we present the main results of our experiments, demonstrating the superior performance of the proposed Preference Optimization (PO) algorithm for COPs. We aim to answer the following questions: 1. How does PO compare to prior RL algorithms on standard benchmarks such as the Traveling Salesman Problem (TSP), the Capacitated Vehicle Routing Problem (CVRP) and the Flexible Flow Shop Problem (FFSP)? 2. How efficiently does PO balance exploitation and exploration by considering entropy, in comparison to traditional RL methods?

**Benchmark Setup.** We implement the PO algorithm across various models, emphasizing that it is a strategy optimization method not tied to a specific model structure, but rather reliant on sampling multiple solutions from identical instances for qualitative comparisons. The fundamental COPs, such as TSP and CVRP, serve as our testbed. In these problems, the reward model $r(x, \tau)$ is defined as the Euclidean length (Len.) of the trajectory $\tau$. The TSP aims to find a Hamiltonian cycle on a graph, minimizing the total trajectory length, while the CVRP incorporates capacity constraints for vehicles and points, along with a depot as the starting point. Our main experiments utilize problems with uniform distribution and 100 nodes, as prescribed in Kool et al. (2019); Kwon et al. (2020). The experiments on the FFSP are conducted to schedule tasks across multiple stages of machines with the objective of minimizing the makespan (MS), which refers to the total time required for completing all tasks. These experiments build upon the model structure proposed by Kwon et al. (2021). Most settings in the model follow the original work, with the exception of the training objective for PO. Further hyper-parameters settings can be found in the Appendix E.2. Most of experiments are conducted on an NVIDIA 24G-RTX 3090 GPU and an Intel Xeon Gold 6133 CPU. Additional experiments on large scale TSP with DIMES Qiu et al. (2022) are included in Appendix F.2.

**Baselines.** We employ well-established heuristic solvers, including LKH3 Helsgaun (2017), HGS Vidal (2022), Concorde Applegate et al. (2006) for routing problems and CPLEX Cplex (2009) for FFSP, to evaluate the optimality gap. The baselines also include notable end-to-end neural solvers for TSP and CVRP: AM Kool et al. (2019), POMO Kwon et al. (2020), Sym-POMO Kim et al. (2022), and Pointerformer Jin et al. (2023): (1) AM utilizes the encoder-decoder architecture from transformers, where the encoder embeds each point in the graph into a vector using multi-head attention, and the decoder generates the trajectory $\tau$ by recursively masking selected points. (2) POMO applies a more efficient training process by imposing diverse starting points for different trajectories and processing them in parallel. For inference, a data-augmentation technique is adopted for exploiting the equivalence of COPs. (3) Sym-NCO considers the symmetry of instances and

Table 1: Experiment results on TSP and CVRP. The result of Len. and Gap are average on 10k instances and the Time are summed of processing 10k instances.

| | Method | TSP (N=100) | | | CVRP (N=100) | | |
|---|---|---|---|---|---|---|---|
| | | Len. ↓ | Gap | Time | Len. ↓ | Gap | Time |
| Heuristic | Concorde | 7.759 | 0.0% | 1.2h | - | - | - |
| | LKH3 | 7.759 | 0.0% | 15.6m | 15.603 | 0.55% | 4h |
| | HGS | - | - | - | 15.518 | 0.0% | 3h |
| Neural Solvers | AM (RL) | 8.023 | 3.40% | 2s | 16.711 | 7.69% | 3s |
| | AM (PO) | 7.981 | 2.86% | 2s | 16.576 | 6.82% | 3s |
| | Pointerformer (RL) | 7.770 | 0.15% | 1m | - | - | - |
| | Pointerformer (PO) | 7.763 | 0.06% | 1m | - | - | - |
| | Sym-NCO (RL) | 7.787 | 0.39% | 10s | 15.768 | 1.59% | 16s |
| | Sym-NCO (PO) | 7.764 | 0.07% | 10s | 15.735 | 1.40% | 16s |
| | POMO (RL) | 7.770 | 0.15% | 1m | 15.791 | 1.76% | 3.3m |
| | POMO (PO) | 7.764 | 0.07% | 1m | 15.730 | 1.37% | 3.3m |
| | POMO (Fine-tuned) | **7.761** | **0.03%** | 1m | **15.703** | **1.19%** | 3.3m |

solutions to enhance the model's solving capability during training; we use its POMO version in our experiments. (4) Pointerformer adopts a more efficient attention module and normalizes the advantages to achieve stable reward signals. We adopt MatNet Kwon et al. (2021) for FFSP.

## 4.1 COMPARISON WITH PRIOR RL ALGORITHMS ON STANDARD BENCHMARKS

We compare the proposed Preference Optimization (PO) method with traditional REINFORCE (RL) methods using the identical model architectures, considering sample efficiency during training, solution quality during inference, and generalization ability (included in Appendix F.1).

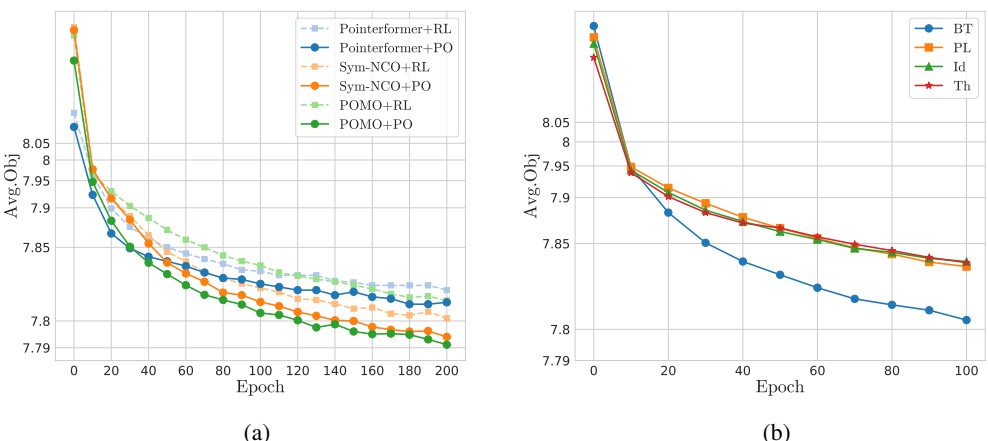

(a)                                                                 (b)

Figure 1: (a) Validation of model performance over epochs for PO (using the Bradley-Terry model) and REINFORCE on TSP100, comparing three different models: Pointerformer, Sym-NCO, and POMO. (b) Comparison of different preference models (Bradley-Terry, Plackett-Luce, Identity, and Thurstone) within PO on TSP100.

**Sample Efficiency.** The training performance of PO and REINFORCE on the POMO, Sym-NCO, and Pointerformer models is compared in terms of sample efficiency. As depicted in Figure 1a, despite employing identical network structures, PO achieves a **convergence speed** 1.5x to 3x faster than REINFORCE on such models. Notably for POMO, training with PO for 60 epochs yields comparable performance to training with RL for 200 epochs. Similar enhancements are observed for Sym-NCO and Pointerformer. This demonstrates the effective acceleration of the training process by PO, resulting in superior performance within fewer training epochs.

Table 2: Experiment results on FFSP. The result of MS and Gap are average on 1k instances and Time are summed of processing 1k instances. * indicate the results are sourced from original paper.

| | Method | FFSP20 | | | FFSP50 | | | FFSP100 | | |
| | | MS. ↓ | Gap | Time | MS. ↓ | Gap | Time | MS. ↓ | Gap | Time |
|---|---|---|---|---|---|---|---|---|---|---|
| Heuristic | CPLEX (60s)* | 46.4 | 84.13% | 17h | | × | | | × | |
| | CPLEX (600s)* | 36.6 | 45.24% | 167h | | × | | | × | |
| | Random | 47.8 | 89.68% | 1m | 93.2 | 88.28% | 2m | 167.2 | 87.42% | 3m |
| | Shortest Job First | 31.3 | 24.21% | 40s | 57.0 | 15.15% | 1m | 99.3 | 11.33% | 2m |
| | Genetic Algorithm | 30.6 | 21.43% | 7h | 56.4 | 13.94% | 16h | 98.7 | 10.65% | 29h |
| | Particle Swarm Opt. | 29.1 | 15.48% | 13h | 55.1 | 11.31% | 26h | 97.3 | 9.09% | 48h |
| Neural Solver | MatNet (RL) | 27.3 | 8.33% | 8s | 51.5 | 4.04% | 14s | 91.5 | 2.58% | 27s |
| | MatNet (RL+Aug) | 25.4 | 0.79% | 3m | 49.6 | 0.20% | 8m | 89.7 | 0.56% | 23m |
| | MatNet (PO) | 27.0 | 7.14% | 8s | 51.3 | 3.64% | 14s | 91.1 | 2.13% | 27s |
| | MatNet (PO+Aug) | **25.2** | - | 3m | **49.5** | - | 8m | **89.2** | - | 23m |

For large-scale TSP using DIMES and FFSP using MatNet, PO achieves comparable performance at only 60%−70% training epochs to that of REINFORCE. Unlike REINFORCE, which converges to suboptimal policies, PO continues to refine and achieve superior solving strategies, demonstrating faster convergence and higher solution quality.

**Solution Quality.** As shown in Table 1, while sharing the same inference times, models trained with PO outperform those trained with the RL objective in terms of solution quality. We also perform 100 epochs of fine-tuning POMO with Local Search (2-opt Croes (1958) for TSP and swap* Vidal (2022) for CVRP) as mentioned in Section 3.4. Interestingly, this approach achieves an optimality gap of only 0.03% on TSP and 1. 19% on CVRP, demonstrating that when approaching the optimal solution, PO can further enhance the policy by using expert knowledge to fine-tune. Moreover, we extended our evaluation to the FFSP. As summarized in Table 2, models trained with PO consistently achieve lower MS and gap compared to their RL counterparts and heuristic solvers. These results confirm that PO not only improves training efficiency but also leads to higher-quality solutions.

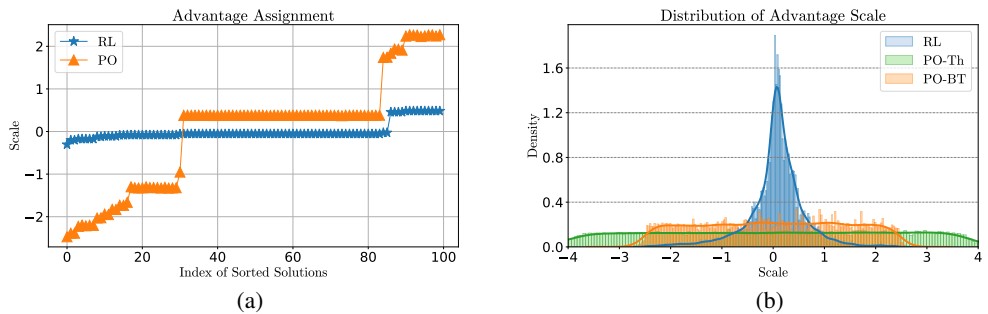

(a)            (b)

Figure 2: (a) Advantage values for 100 solutions sampled from the trained POMO model, where sorting highlights the advantage assignment patterns. The horizontal lines at different scales indicate that Kwon et al. (2020) can lead to similar cycles, resulting in similar advantage values. (b) Distribution of advantage scales for 50,000 sampled solutions, comparing REINFORCE, PO with the Thurstone model (PO-Th), and PO with the Bradley-Terry model (PO-BT).

## 4.2 HOW EFFICIENTLY DOES PO BALANCE EXPLOITATION AND EXPLORATION?

**Consistency of Policy.** A key advantage of the proposed PO method is its ability to consistently emphasize better solutions, independent of the numerical values of the advantage function. Figure 2a compares the advantage assignment between PO and the conventional REINFORCE algorithm. PO effectively separates high-quality trajectories by assigning them positive advantage values while allocating negative values to low-quality ones. In contrast, REINFORCE struggles to differentiate trajectory quality, with most advantage values centered around zero. This distinction showcases

PO's capability to both highlight superior solutions and suppress inferior ones, leading to more efficient exploration and faster convergence. Additionally, Figure 2b presents the distribution of advantage scales, where RL exhibits a narrow, peaked distribution around zero, indicating limited differentiation. Conversely, PO-based methods display broader distributions, covering a wider range of both positive and negative values. This indicates PO's enhanced ability to distinguish between high- and low-quality trajectories, further supporting its effectiveness in policy optimization.

Furthermore, Figure 3a evaluates the consistency of the policies. PO significantly improves the consistency of the learned policies compared to REINFORCE, and fine-tuning with local search further enhances this consistency.

**Diversity for Exploration.** One limitation of the REINFORCE algorithm is its incompatibility with entropy regularization at the trajectory level. In contrast, the PO method is derived from an entropy-regularized objective, which inherently promotes exploration. We compare the sum of entropy at each step in the trajectory during the early training phase between PO and REINFORCE. As shown in Figure 3b, the model trained using PO achieves significantly higher entropy, indicating a

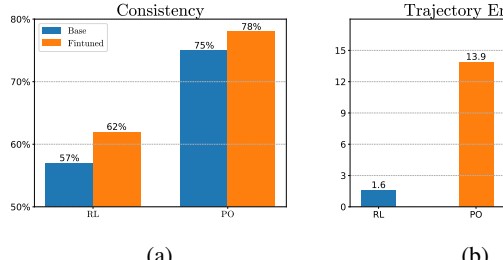

(a)        (b)

Figure 3: (a) Consistency measured as $p(\pi(\tau_1) > \pi(\tau_2) \mid r(\tau_1) > r(\tau_2))$, evaluated on the trained POMO model. PO shows higher consistency than RL, with further improvement after fine-tuning. (b) Trajectory entropy, calculated as the sum of entropy at each step, compared across models. The values are measured during early training for RL and PO, and during the initial phase of fine-tuning for PO+LS, indicating higher exploration in PO and PO+LS compared to RL.

more diverse set of explored strategies. On the other hand, the RL update scheme results in lower entropy, potentially leading to less efficient exploration. Additionally, using PO to fine-tune a trained model with local search, which integrates external expert knowledge, further enhances strategy diversity. In conclusion, PO effectively balances exploration and exploitation, enabling the model to explore the solution space more thoroughly.

**Study on Preference Models.** A crucial aspect of PO is the choice of the preference model, as discussed in Section 3.3. Different preference models may lead to varying implicit reward models, as outlined in Eq. 7 and 8. Assuming a differentiable paired preference model $f(\cdot)$, the generalized form of the latent reward assigned for each $\tau$ will be: $\frac{1}{|S_x|} \sum_{\tau' \in S_x} \left[ g_f(\tau, \tau', x) - g_f(\tau', \tau, x) \right]$,

where $g_f(\tau, \tau', x) = \mathbf{1}_{[r(x,\tau)>r(x,\tau')]} \cdot \frac{f'\left(\hat{r}_\theta(x,\tau) - \hat{r}_\theta(x,\tau')\right)}{f\left(\hat{r}_\theta(x,\tau) - \hat{r}_\theta(x,\tau')\right)}$ for any $\tau' \in S_x$. The results, shown in Figure 1b, indicate that the Bradley-Terry model consistently outperforms the others in terms of convergence on TSP. This suggests an interesting direction for further research, exploring the relationships among these preference models and their impact on the optimization landscape.

## 5 CONCLUSION

In this paper, we introduced **Preference Optimization**, a novel framework for solving COPs. By transforming quantitative reward signals into qualitative preference signals, PO addresses the challenges of diminishing reward differences and inefficient exploration inherent in traditional RL approaches. We enhanced PO by integrating heuristic local search techniques into the fine-tuning process, enabling neural solvers to generate near-optimal solutions without additional inference time. Extensive experimental results demonstrate the practical viability and effectiveness of our approach, achieving superior sample efficiency and solution quality compared to traditional RL algorithms.

While PO shows significant promise, the stability of the reparameterized reward function across different COPs requires further investigation. Looking ahead, applying PO to optimization problems where reward signals are difficult to design but preference information is readily available, such as multi-objective optimization, remains a valuable direction.

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

# A ILLUSTRATION OF THE PO FRAMEWORK

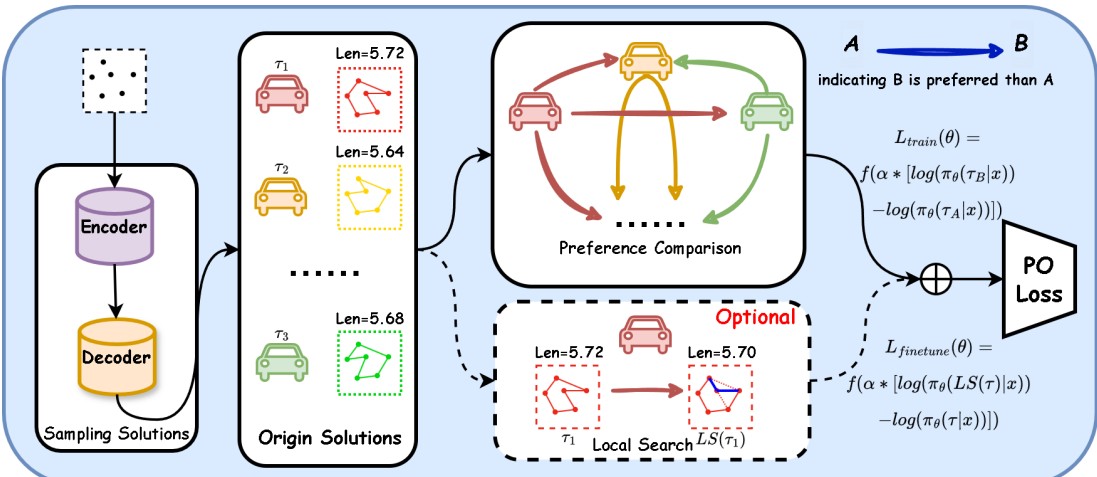

Figure 4: **Framework of the Preference Optimization (PO) Algorithm.** This figure illustrates the workflow of the PO algorithm. The process starts with the parameterized **Encoder-Decoder module** (left), which samples solution trajectories ($\tau_1, \tau_2, \ldots$) for a given COP, forming the **Origin Solutions**. In the **Preference Comparison module** (center), pairwise comparisons are conducted between solutions based on their performance (e.g., trajectory length). The arrows indicate preference relationships (e.g., when $len(B) < len(A)$, $B$ is preferred over $A$), and these preferences are used to compute the PO Loss. The **Optional Local Search step** (bottom) refines selected solutions ($\tau$) by applying search techniques (e.g., 2-Opt), producing improved solutions ($LS(\tau)$). These refined solutions contribute additional gradient signals ($L_{\text{finetune}}$) during the fine-tuning stage. This framework illustrate how PO transforms quantitative rewards into qualitative preferences, ensuring robust training with or without local search.

# B COMBINATORIAL OPTIMIZATION PROBLEMS: TSP AND CVRP

We provide concise introductions to two fundamental combinatorial optimization problems: the Traveling Salesman Problem (TSP) and the Capacitated Vehicle Routing Problem (CVRP).

## B.1 TRAVELING SALESMAN PROBLEM

The Traveling Salesman Problem (TSP) seeks to determine the shortest possible route that visits each city exactly once and returns to the origin city. Formally, given a set of cities $\mathcal{C} = \{c_1, c_2, \ldots, c_n\}$ and a distance matrix $D$ where $D_{i,j}$ represents the distance between cities $c_i$ and $c_j$, the objective is to find a trajectory $\tau = (c_1, c_2, \ldots, c_n, c_1)$ that minimizes the total travel distance:

$$\min_{\tau} \sum_{k=1}^{n} D_{\tau(k),\tau(k+1)}.$$

Subject to:

$$\tau \text{ is a permutation of } \mathcal{C}, \quad \tau(n+1) = \tau(1).$$

Here, $\tau(k)$ denotes the $k$-th city in the trajectory, and the constraint $\tau(n+1) = \tau(1)$ ensures that the tour returns to the starting city.

## B.2 CAPACITATED VEHICLE ROUTING PROBLEM

The Capacitated Vehicle Routing Problem (CVRP) extends the TSP by introducing multiple vehicles with limited carrying capacities. The goal is to determine the optimal set of routes for a fleet of vehicles to deliver goods to a set of customers, minimizing the total distance traveled while respecting the capacity constraints of the vehicles.

Formally, given:

- A depot $c_0$,

- A set of customers $\mathcal{C} = \{c_1, c_2, \ldots, c_n\}$,

- A demand $d_i$ for each customer $c_i$,

- A distance matrix $D$ where $D_{i,j}$ represents the distance between locations $c_i$ and $c_j$,

- A fleet of $m$ identical vehicles, each with capacity $Q$,

the objective is to assign trajectories $\{\tau_1, \tau_2, \ldots, \tau_m\}$ to the vehicles such that each customer is visited exactly once, the total demand on any trajectory does not exceed the vehicle capacity $Q$, and the total distance traveled by all vehicles is minimized:

$$\min_{\{\tau_1, \tau_2, \ldots, \tau_m\}} \sum_{k=1}^{m} \sum_{l=1}^{|\tau_k|-1} D_{\tau_k(l), \tau_k(l+1)}.$$

Subject to:

$$\tau_k(1) = \tau_k(|\tau_k|) = c_0, \quad \forall k \in \{1, 2, \ldots, m\},$$

$$\bigcup_{k=1}^{m} \{\tau_k(2), \tau_k(3), \ldots, \tau_k(|\tau_k| - 1)\} = \mathcal{C},$$

$$\tau_k(i) \neq \tau_k(j) \quad \forall k \in \{1, 2, \ldots, m\}, \forall i \neq j,$$

$$\sum_{c_i \in \tau_k} d_i \leq Q, \quad \forall k \in \{1, 2, \ldots, m\}.$$

Here, $\tau_k(l)$ denotes the $l$-th location in the trajectory $\tau_k$ assigned to vehicle $k$. The constraints ensure that:

- Each trajectory starts and ends at the depot $c_0$.

- Every customer is visited exactly once across all trajectories.

- No customer is visited more than once within the same trajectory.

- The total demand served by each vehicle does not exceed its capacity $Q$.

### B.3 TRAJECTORY REPRESENTATION

In both TSP and CVRP, a trajectory $\tau$ represents a sequence of actions or decisions made by the policy to construct a solution. For TSP, $\tau$ is a single cyclic permutation of the cities, whereas for CVRP, $\tau$ comprises multiple routes, each assigned to a vehicle. Our Preference Optimization framework utilizes these trajectories to model and compare solution quality through preference signals derived from statistical comparison models.

### B.4 FLEXIBLE FLOW SHOP PROBLEM

The Flexible Flow Shop Problem (FFSP) is a combinatorial optimization problem commonly encountered in scheduling tasks. It generalizes the classic flow shop problem by allowing multiple parallel machines at each stage, where jobs can be processed on any machine within a stage. The primary goal is to assign and sequence jobs across stages to minimize the makespan, which is the total time required to complete all jobs.

The optimization objective for FFSP can be mathematically formulated as:

$$\min_{\sigma, \mathbf{x}} C_{\max} = \max_{j \in \mathcal{J}} \left\{ C_j^{m_s} \right\},$$

subject to:

$$C_j^{m_s} = S_j^{m_s} + p_j^{m_s}, \quad \forall j \in \mathcal{J}, \forall m_s \in \mathcal{M},$$
$$S_j^{m_s} \geq C_j^{m_{s-1}}, \quad \forall j \in \mathcal{J}, \forall m_{s-1} \in \mathcal{M},$$
$$S_j^{m_s} \geq C_{j'}^{m_s}, \quad \forall (j, j') \in \mathcal{J}, \text{ if } \sigma(j) > \sigma(j'),$$
$$x_{j,m_s} = 1, \quad \text{if job } j \text{ is assigned to machine } m_s,$$
$$\sum_{m_s \in \mathcal{M}} x_{j,m_s} = 1, \quad \forall j \in \mathcal{J}.$$

Here: $\mathcal{J}$ is the set of jobs. $\mathcal{M}$ is the set of machines at each stage. $\sigma$ represents the sequence of jobs. $x$ is the assignment matrix of jobs to machines. $S_j^{m_s}$ is the start time of job $j$ on machine $m_s$. $C_j^{m_s}$ is the completion time of job $j$ on machine $m_s$. $p_j^{m_s}$ is the processing time of job $j$ on machine $m_s$. $C_{\max}$ is the makespan to be minimized.

The constraints ensure that jobs are scheduled sequentially on machines, maintain precedence, and adhere to the assignment rules. The FFSP is NP-hard and challenging to solve for large-scale instances.

## C    PREFERENCE MODELS

In this section, we provide a concise overview of three widely used preference models: the Bradley-Terry (BT) model, the Thurstone model, and the Plackett-Luce (PL) model. These models are fundamental in statistical comparison modeling and form the basis for transforming quantitative reward signals into qualitative preference signals in our Preference Optimization (PO) framework.

### C.1    BRADLEY-TERRY MODEL

The Bradley-Terry model is a probabilistic model used for pairwise comparisons. It assigns a positive parameter to each trajectory $\tau_i$, representing its preference strength. The probability that trajectory $\tau_i$ is preferred over trajectory $\tau_j$ is given by:

$$p(\tau_i \succ \tau_j) = \frac{\exp(\hat{r}(\tau_i))}{\exp(\hat{r}(\tau_i)) + \exp(\hat{r}(\tau_j))}$$
$$= \frac{1}{1 + \exp(-(\hat{r}(\tau_i) - \hat{r}(\tau_j)))}$$
$$= \sigma(\hat{r}(\tau_i) - \hat{r}(\tau_j)).$$

This model assumes that the preference between any two trajectories depends solely on their respective preference strengths, and it maintains the property of transitivity.

### C.2    THURSTONE MODEL

The Thurstone model, also known as the Thurstone-Mosteller model, is based on the assumption that each trajectory $\tau_i$ has an associated latent score $s_i$, which is normally distributed. The probability that trajectory $\tau_i$ is preferred over trajectory $\tau_j$ is modeled as:

$$p(\tau_i \succ \tau_j) = \Phi\left(\frac{\hat{r}(\tau_i) - \hat{r}(\tau_j)}{\sigma}\right),$$

where $\Phi$ is the cumulative distribution function of the standard normal distribution, and $\sigma$ represents the standard deviation of the underlying noise. This model accounts for uncertainty in preferences and allows for probabilistic interpretation of comparisons. We adopt a normal distribution throughout this work.

## C.3 PLACKETT-LUCE MODEL

The Plackett-Luce model extends pairwise comparisons to handle full rankings of multiple trajectories. It assigns a positive parameter $\lambda_i$ to each trajectory $\tau_i$, representing its utility. Given a set of trajectories to be ranked, the probability of observing a particular ranking $\tau = (\tau_1, \tau_2, \ldots, \tau_n)$ is given by:

$$P(\tau) = \prod_{k=1}^{n} \frac{\exp(\hat{r}(\tau_k))}{\sum_{j=k}^{n} \exp(\hat{r}(\tau_j))}.$$

This model is particularly useful for modeling complete rankings and can be extended to partial rankings. It preserves the property of independence of irrelevant alternatives and allows for flexible representation of preferences over multiple trajectories.

# D MATHEMATICAL DERIVATIONS

## D.1 DERIVING THE OPTIMAL POLICY FOR ENTROPY-REGULARIZED RL OBJECTIVE

In this section, we derive the analytical solution for the optimal policy in an entropy-regularized reinforcement learning objective.

Starting from the entropy-regularized RL objective in Eq. 3:

$$\max_{\pi} \mathbb{E}_{x \sim \mathcal{D}, \ \tau \sim \pi(\tau|x)} \left[ r(x, \tau) \right] + \alpha \, \mathcal{H} \left( \pi(\tau \mid x) \right),$$

where $\mathcal{H} \left( \pi(\tau \mid x) \right) = -\mathbb{E}_{\tau \sim \pi(\tau|x)} \left( \log \pi(\tau \mid x) \right)$ is the entropy of the policy, and $\alpha > 0$ is the regularization coefficient.

We can rewrite the objective as:

$$\max_{\pi} \mathbb{E}_{x \sim \mathcal{D}, \ \tau \sim \pi(\tau|x)} \left[ r(x, \tau) - \alpha \log \pi(\tau \mid x) \right]. \tag{10}$$

Our goal is to find the policy $\pi^*(\tau \mid x)$ that maximizes this objective. To facilitate the derivation, we can express the problem as a minimization:

$$\min_{\pi} \mathbb{E}_{x \sim \mathcal{D}, \ \tau \sim \pi(\tau|x)} \left[ \log \pi(\tau \mid x) - \frac{1}{\alpha} r(x, \tau) \right]. \tag{11}$$

Notice that:

$$\log \pi(\tau \mid x) - \frac{1}{\alpha} r(x, \tau) = \log \frac{\pi(\tau \mid x)}{\exp \left( \frac{1}{\alpha} r(x, \tau) \right)}. \tag{12}$$

Introduce the partition function $Z(x) = \sum_{\tau} \exp \left( \frac{1}{\alpha} r(x, \tau) \right)$, and define the probability distribution:

$$\pi^*(\tau \mid x) = \frac{1}{Z(x)} \exp \left( \frac{1}{\alpha} r(x, \tau) \right). \tag{13}$$

This defines a valid probability distribution over trajectories $\tau$ for each instance $x$, as $\pi^*(\tau \mid x) > 0$ and $\sum_{\tau} \pi^*(\tau \mid x) = 1$.

Substituting Eq. equation 13 into Eq. equation 12, we have:

$$\log \pi(\tau \mid x) - \frac{1}{\alpha} r(x, \tau) = \log \frac{\pi(\tau \mid x)}{\pi^*(\tau \mid x)} + \log Z(x). \tag{14}$$

Therefore, the minimization problem in Eq. equation 11 becomes:

$$\min_{\pi} \mathbb{E}_{x \sim \mathcal{D}} \left[ \mathbb{E}_{\tau \sim \pi(\tau|x)} \left[ \log \frac{\pi(\tau \mid x)}{\pi^*(\tau \mid x)} \right] + \log Z(x) \right]. \tag{15}$$

Since $\log Z(x)$ does not depend on $\pi$, minimizing over $\pi$ reduces to minimizing the Kullback-Leibler (KL) divergence between $\pi(\tau \mid x)$ and $\pi^*(\tau \mid x)$:

$$\min_{\pi} \mathbb{E}_{x \sim \mathcal{D}} \left[ D_{\mathrm{KL}} \left( \pi(\tau \mid x) \parallel \pi^*(\tau \mid x) \right) \right], \tag{16}$$

where the KL divergence is defined as:

$$D_{\mathrm{KL}} \left( \pi(\tau \mid x) \parallel \pi^*(\tau \mid x) \right) = \mathbb{E}_{\tau \sim \pi(\tau|x)} \left[ \log \frac{\pi(\tau \mid x)}{\pi^*(\tau \mid x)} \right].$$

The KL divergence is minimized when $\pi(\tau \mid x) = \pi^*(\tau \mid x)$ almost everywhere. Therefore, the optimal policy is:

$$\pi^*(\tau \mid x) = \frac{1}{Z(x)} \exp \left( \frac{1}{\alpha} r(x, \tau) \right). \tag{17}$$

This shows that the optimal policy under the entropy-regularized RL objective is proportional to the exponentiated reward function, normalized by the partition function $Z(x)$.

**Conclusion.** We have derived that the optimal policy $\pi^*(\tau \mid x)$ in the entropy-regularized RL framework is given by Eq. equation 17. This policy assigns higher probabilities to trajectories with higher rewards, balanced by the entropy regularization parameter $\alpha$, which controls the trade-off between exploitation and exploration.

### D.2 Proof of Proposition 1

**Proposition 2** *Let $\hat{r}(x, \tau)$ be a reward function consistent with the Bradley-Terry, Thurstone, or Plackett-Luce models. For a given reward function $\hat{r}'(x, \tau)$, if there exists a function $h(x)$ such that $\hat{r}'(x, \tau) = \hat{r}(x, \tau) - h(x)$, then both $\hat{r}(x, \tau)$ and $\hat{r}'(x, \tau)$ induce the same optimal policy in the context of an entropy-regularized reinforcement learning problem.*

*Proof:* In an entropy-regularized reinforcement learning framework, the optimal policy $\pi^*(\tau \mid x)$ for a given reward function $\hat{r}(x, \tau)$ is given by:

$$\pi^*(\tau \mid x) = \frac{1}{Z(x)} \exp \left( \frac{1}{\alpha} \hat{r}(x, \tau) \right),$$

where $\alpha > 0$ is the temperature parameter (inverse of the regularization coefficient), and $Z(x)$ is the partition function defined as:

$$Z(x) = \sum_{\tau} \exp \left( \frac{1}{\alpha} \hat{r}(x, \tau) \right).$$

Similarly, for the reward function $\hat{r}'(x, \tau) = \hat{r}(x, \tau) - h(x)$, the optimal policy $\pi'^*(\tau \mid x)$ is:

$$\pi'^*(\tau \mid x) = \frac{1}{Z'(x)} \exp \left( \frac{1}{\alpha} \hat{r}'(x, \tau) \right) = \frac{1}{Z'(x)} \exp \left( \frac{1}{\alpha} [\hat{r}(x, \tau) - h(x)] \right), \tag{18}$$

where $Z'(x)$ is the partition function corresponding to $\hat{r}'(x, \tau)$:

$$Z'(x) = \sum_{\tau} \exp \left( \frac{1}{\alpha} \hat{r}'(x, \tau) \right) = \sum_{\tau} \exp \left( \frac{1}{\alpha} [\hat{r}(x, \tau) - h(x)] \right).$$

Simplifying the exponent in Equation equation 18:

$$\exp\left(\frac{1}{\alpha}[\hat{r}(x,\tau) - h(x)]\right) = \exp\left(\frac{1}{\alpha}\hat{r}(x,\tau)\right)\exp\left(-\frac{1}{\alpha}h(x)\right).$$

Since $h(x)$ depends only on $x$ and not on $\tau$, the term $\exp\left(-\frac{1}{\alpha}h(x)\right)$ is a constant with respect to $\tau$. Therefore, we can rewrite Equation equation 18 as:

$$\pi'^*(\tau \mid x) = \frac{1}{Z'(x)}\exp\left(-\frac{1}{\alpha}h(x)\right)\exp\left(\frac{1}{\alpha}\hat{r}(x,\tau)\right). \tag{19}$$

Combining constants:

$$\pi'^*(\tau \mid x) = \left(\frac{\exp\left(-\frac{1}{\alpha}h(x)\right)}{Z'(x)}\right)\exp\left(\frac{1}{\alpha}\hat{r}(x,\tau)\right).$$

Notice that the term $\frac{\exp\left(-\frac{1}{\alpha}h(x)\right)}{Z'(x)}$ is a normalization constant that ensures $\sum_\tau \pi'^*(\tau \mid x) = 1$. Similarly, for $\pi^*(\tau \mid x)$, the normalization constant is $\frac{1}{Z(x)}$.

Since both $\pi^*(\tau \mid x)$ and $\pi'^*(\tau \mid x)$ are proportional to $\exp\left(\frac{1}{\alpha}\hat{r}(x,\tau)\right)$, they differ only by their respective normalization constants. Therefore, they assign the same relative probabilities to trajectories $\tau$.

To formalize this, consider any two trajectories $\tau_1$ and $\tau_2$. The ratio of their probabilities under $\pi^*(\tau \mid x)$ is:

$$\frac{\pi^*(\tau_1 \mid x)}{\pi^*(\tau_2 \mid x)} = \frac{\exp\left(\frac{1}{\alpha}\hat{r}(x,\tau_1)\right)}{\exp\left(\frac{1}{\alpha}\hat{r}(x,\tau_2)\right)} = \exp\left(\frac{1}{\alpha}[\hat{r}(x,\tau_1) - \hat{r}(x,\tau_2)]\right). \tag{20}$$

Similarly, under $\pi'^*(\tau \mid x)$:

$$\frac{\pi'^*(\tau_1 \mid x)}{\pi'^*(\tau_2 \mid x)} = \frac{\exp\left(\frac{1}{\alpha}\hat{r}'(x,\tau_1)\right)}{\exp\left(\frac{1}{\alpha}\hat{r}'(x,\tau_2)\right)} = \exp\left(\frac{1}{\alpha}[\hat{r}'(x,\tau_1) - \hat{r}'(x,\tau_2)]\right). \tag{21}$$

Substituting $\hat{r}'(x,\tau) = \hat{r}(x,\tau) - h(x)$:

$$\hat{r}'(x,\tau_1) - \hat{r}'(x,\tau_2) = [\hat{r}(x,\tau_1) - h(x)] - [\hat{r}(x,\tau_2) - h(x)] = \hat{r}(x,\tau_1) - \hat{r}(x,\tau_2).$$

Therefore, the ratios in Equations equation 20 and equation 21 are equal:

$$\frac{\pi^*(\tau_1 \mid x)}{\pi^*(\tau_2 \mid x)} = \frac{\pi'^*(\tau_1 \mid x)}{\pi'^*(\tau_2 \mid x)}.$$

Since the policies assign the same relative probabilities to all trajectories, and they are both properly normalized, it follows that:

$$\pi^*(\tau \mid x) = \pi'^*(\tau \mid x), \quad \forall \tau.$$

Thus, $\hat{r}(x,\tau)$ and $\hat{r}'(x,\tau)$ induce the same optimal policy in the context of an entropy-regularized reinforcement learning problem.

This result holds for the Bradley-Terry, Thurstone, and Plackett-Luce models because these models relate preferences to differences in reward values, and any constant shift $h(x)$ in the reward function does not affect the differences between reward values for different trajectories.

# E  EXPERIMENT DETAIL AND SETTING

## E.1  IMPLEMENTATION DETAILS OF THE CODE

The implementation of the Preference Optimization (PO) algorithm in Python using PyTorch is as follows:

```python
import torch.nn.functional as F
def preference_optimazation(reward, log_prob):
    """
        reward: reward for all solutions, shape(B, P)
        log_prob: policy log prob, shape(B, P)
    """
    preference = reward[:, :, None] > reward[:, None, :]
    log_prob_pair = log_prob[:, :, None] - log_prob[:, None, :]
    pf_log = torch.log(F.sigmoid(self.alpha * log_prob_pair))
    loss = -torch.mean(pf_log * preference)

    return loss
```

## E.2  HYPERPARAMETER SETTING

In our experimental setup, we set the tanh clip to 50, which has been shown to facilitate the training process Jin et al. (2023). The following table presents the parameter settings for the four training frameworks: POMO Kwon et al. (2020), Pointerformer Jin et al. (2023), AM Kool et al. (2019), and Sym-NCO Kim et al. (2023).

**POMO** framework hyperparameter settings:

Table 3: Hyperparameter setting for POMO.

|  | TSP | CVRP |
|---|---|---|
| Alpha | 0.05 | 0.03 |
| Preference Function | BT | BT |
| Epochs | 2000 | 4000 |
| Epochs (Finetune) | 100 | 200 |
| Epoch Size | 100000 | 50000 |
| Encoder Layer Number | 6 | 6 |
| Batch Size | 64 | 64 |
| Embedding Dimension | 128 | 128 |
| Attention Head Number | 8 | 8 |
| Feed Forward Dimension | 512 | 512 |
| Tanh Clip | 50 | 50 |
| Learning Rate | 3e-4 | 3e-4 |

**Pointerformer** framework hyperparameter settings:

Table 4: Hyperparameter setting for Pointerformer.

|  | TSP |
|---|---|
| Alpha | 0.05 |
| Preference Function | BT |
| Epochs | 2000 |
| Epoch Size | 100000 |
| Batch Size | 64 |
| Embedding Dimension | 128 |
| Attention Head Number | 8 |
| Feed Forward Dimension | 512 |
| Encoder Layer Number | 6 |
| Learning Rate | 1e-4 |

**AM** framework hyperparameter settings. Batch size of 256 contains 16 instances, each with 16 solutions, totaling 256 trajectories:

Table 5: Hyperparameter setting for AM.

|  | TSP | CVRP |
|---|---|---|
| Alpha | 0.05 | 0.03 |
| Preference Function | BT | BT |
| Epochs | 100 | 100 |
| Epoch Size | 1280000 | 1280000 |
| Encoder Layer Number | 3 | 3 |
| Batch Size | 256 | 256 |
| Embedding Dimension | 128 | 128 |
| Attention Head Number | 8 | 8 |
| Tanh Clip | 50 | 50 |
| Learning Rate | 1e-4 | 1e-4 |

**Sym-NCO** framework hyperparameter settings:

Table 6: Hyperparameter setting for Sym-NCO.

|  | TSP | CVRP |
|---|---|---|
| Alpha | 0.05 | 0.03 |
| Preference Function | BT | BT |
| Epochs | 2000 | 4000 |
| Epoch Size | 100000 | 50000 |
| Batch Size | 64 | 64 |
| SR Size | 2 | 2 |
| Embedding Dimension | 128 | 128 |
| Attention Head Number | 8 | 8 |
| Feed Forward Dimension | 512 | 512 |
| Encoder Layer Number | 6 | 6 |
| Learning Rate | 1e-4 | 1e-4 |

**DIMES** framework hyperparameter settings:

Table 7: Hyperparameter Setting for DIMES.

|  | TSP500 | TSP1000 | TSP10000 |
|---|---|---|---|
| Alpha | 2 | 2 | 2 |
| Preference Function | Identity | Identity | Identity |
| KNN K | 50 | 50 | 50 |
| Outer Opt | AdamW | AdamW | AdamW |
| Outer Opt LR | 0.001 | 0.001 | 0.001 |
| Outer Opt WD | 1e-5 | 1e-5 | 1e-5 |
| Net Units | 32 | 32 | 32 |
| Net Act | SiLU | SiLU | SiLU |
| Emb Depth | 12 | 12 | 12 |
| Par Depth | 3 | 3 | 3 |
| Training Batch Size | 3 | 3 | 3 |

**MATNET** framework hyperparameter settings:

Table 8: Hyperparameter Setting for MATNET.

|  | FFSP20 | FFSP50 | FFSP100 |
|---|---|---|---|
| Alpha | 1.5 | 1.5 | 1 |
| Preference Function | Identity | Identity | Identity |
| Pomo Size | 24 | 24 | 24 |
| Epochs | 100 | 150 | 200 |
| Epoch Size | 1000 | 1000 | 1000 |
| Encoder Layer Number | 3 | 3 | 3 |
| Batch Size | 50 | 50 | 50 |
| Embedding Dimension | 256 | 256 | 256 |
| Attention Head Number | 16 | 16 | 16 |
| Feed Forward Dimension | 512 | 512 | 512 |
| Tanh Clip | 10 | 10 | 10 |
| Learning Rate | 1e-4 | 1e-4 | 1e-4 |

### E.3 POMO TRAINING RESULTS

Figure 5 compares the training efficiency of the PO and RL algorithms for TSP and CVRP. In the TSP task (a), PO reaches an objective value of 7.785 at epoch 400, while RL requires up to 1600 epochs to achieve comparable performance, demonstrating the sample efficiency of PO. This difference becomes more pronounced as training progresses. In the more challenging CVRP environment (b), PO continues to outperform RL, indicating its robustness and effectiveness in handling more complex optimization problems.

For TSP, each training epoch takes approximately 9 minutes, while each finetuning epoch with local search takes about 12 minutes. For CVRP, a training epoch takes about 8 minutes, and a finetuning epoch takes around 20 minutes. Since local search is executed on the CPU, it does not introduce additional GPU inference time. The finetuning phase constitutes 5% of the total epochs, adding a manageable overhead to the overall training time.

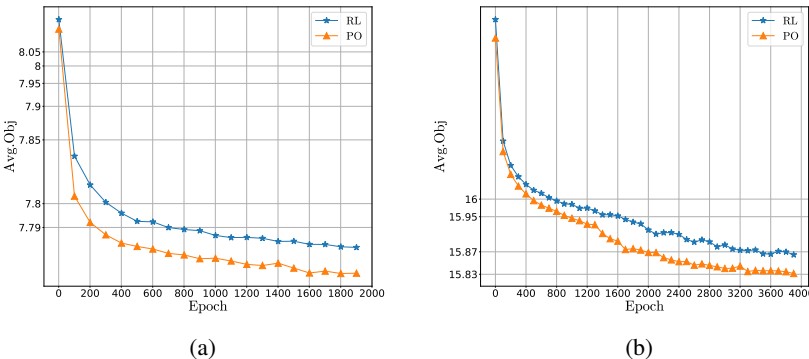

(a)                                             (b)

Figure 5: (a) Training curve for TSP (N=100) over 2000 epochs. (b) Training curve for CVRP (N=100) over 4000 epochs.

## F ADDITIONAL EXPERIMENTS.

### F.1 GENERALIZATION

We conducted a zero-shot cross-distribution evaluation, where models were tested on data from unseen distributions. Since models trained purely with RL tend to overfit to the training data distribution Zhou et al. (2023), they may struggle with different reward functions in new distributions. However, training with PO helps mitigate this overfitting by avoiding the need for ground-truth reward signals. Following the diverse distribution setup in Bi et al. (2022), the results are summarized

in Table 9. Our findings show that the model trained with PO outperforms the original RL-based model across all scenarios.

Table 9: Zero-shot generalization experiment results. The Len and Gap are average on 10k instances.

| | Method | Cluster Len.↓ | Cluster Gap | Expansion Len.↓ | Expansion Gap | Explosion Len.↓ | Explosion Gap | Grid Len.↓ | Grid Gap | Implosion Len.↓ | Implosion Gap |
|---|---|---|---|---|---|---|---|---|---|---|---|
| TSP | LKH | 3.66 | 0.00% | 5.38 | 0.00% | 5.83 | 0.00% | 7.79 | 0.00% | 7.61 | 0.00% |
| | POMO-RL | 3.74 | 2.09% | 5.41 | 0.60% | 5.85 | 0.20% | 7.80 | 0.16% | 7.63 | 0.15% |
| | POMO-PO | **3.70** | **1.12%** | **5.40** | **0.34%** | **5.84** | **0.06%** | **7.79** | **0.04%** | **7.62** | **0.05%** |
| CVRP | HGS | 7.79 | 0.00% | 11.38 | 0.00% | 12.35 | 0.00% | 15.59 | 0.00% | 15.47 | 0.00% |
| | POMO-RL | 7.97 | 2.28% | 11.51 | 1.29% | 12.48 | 0.97% | 15.79 | 0.86% | 15.60 | 0.87% |
| | POMO-PO | **7.93** | **1.73%** | **11.49** | **1.12%** | **12.45** | **0.76%** | **15.76** | **0.63%** | **15.57** | **0.65%** |

## F.2 EXPERIMENTS ON LARGE SCALE PROBLEMS

We further conduct experiments on large-scale TSP problems to validate the effectiveness of PO using the DIMES model Qiu et al. (2022). DIMES leverages a reinforcement learning and meta-learning framework to train a parameterized heatmap, with REINFORCE as the optimization method in their original experiments. Solutions are generated by combining the heatmap with various heuristic methods, such as greedy decoding, MCTS, 2-Opt, or fine-tuning methods like Active Search (AS), which further train the solver for each instance.

As summarized in Table 10, our experiments demonstrate that PO improves the quality of the heatmap representations compared to REINFORCE. Across all decoding strategies (e.g., greedy, sampling, MCTS, AS), PO-trained models consistently outperform their REINFORCE-trained counterparts in terms of solution quality, as evidenced by lower gap percentages across TSP500, TSP1000, and TSP10000. This confirms that PO enhances the learned policy, making it more effective regardless of the heuristic decoding method applied.

Table 10: Experiment results on large scale TSP.

| Method | TSP500 Len. ↓ | TSP500 Gap | TSP500 Time | TSP1000 Len. ↓ | TSP1000 Gap | TSP1000 Time | TSP10000 Len. ↓ | TSP10000 Gap | TSP10000 Time |
|---|---|---|---|---|---|---|---|---|---|
| LKH-3 | 16.55 | 0.00% | 46.28m | 23.12 | 0.00% | 2.57h | 71.79 | 0.00% | 8.8h |
| DIMES-G(RL) | 19.30 | 16.62% | 0.8m | 26.58 | 14.96% | 1.5m | 86.38 | 20.36% | 2.3m |
| DIMES-G(PO) | 18.82 | 13.73% | 0.8m | 26.22 | 13.39% | 1.5m | 85.33 | 18.87% | 2.3m |
| DIMES-S(RL) | 19.11 | 15.47% | 0.9m | 26.37 | 14.05% | 1.8m | 85.79 | 19.50% | 2.4m |
| DIMES-S(PO) | 18.75 | 13.29% | 0.9m | 26.07 | 12.74% | 1.8m | 85.21 | 18.67% | 2.4m |
| DIMES-AS(RL) | 17.82 | 7.68% | 2h | 24.99 | 8.09% | 4.3h | 80.68 | 12.39% | 2.5h |
| DIMES-AS(PO) | 17.78 | 7.42% | 2h | 24.73 | 6.97% | 4.3h | 80.14 | 11.64% | 2.5h |
| DIMES-MCTS(RL) | 16.93 | 2.30% | 3m | 23.96 | 3.65% | 6.3m | 74.83 | 4.24% | 27m |
| **DIMES-MCTS(PO)** | **16.89** | **2.05%** | 3m | **23.96** | **3.65%** | 6.3m | **74.77** | **4.15%** | 27m |

