# OpenReview forum: "Preference Optimization for Combinatorial Optimization Problems"
_ICLR.cc/2025/Conference — ICLR 2025 Conference Withdrawn Submission_

### Official Review · Reviewer_1APy · 2024-10-29

**Soundness:** 3
**Presentation:** 4
**Contribution:** 4
**Rating:** 6
**Confidence:** 4

**Summary:**

This paper proposes a method for training an artificial neural network model to solve combinatorial optimization problems. The authors identify issues with the REINFORCE-based parameter update method, which has been widely used in various existing Neural Combinatorial Optimization studies, and propose Preference Optimization(PO) as a solution. Furthermore, the authors present a method that integrates the proposed PO method with existing local search techniques for training. They apply the proposed method to AM, POMO, Sym-NCO, and Pointerformer, demonstrating better results in TSP and CVRP compared to traditional REINFORCE-based optimization methods. The authors also report strong performance in generalization experiments.

**Strengths:**

- The authors propose Preference Optimization (PO) as an optimization method for Neural Combinatorial Optimization (NCO) models. They demonstrate, through experiments on TSP100 and CVRP100, that applying the proposed PO to existing studies such as AM, POMO, Sym-NCO, and Pointerformer leads to improved model performance.

- The proposed Preference Optimization method in this paper is particularly noteworthy in that it can be applied to various existing NCO studies to enhance their performance. It is expected to be applicable to a wide range of future NCO research.

- In generalization experiments, the model trained with Preference Optimization also outperformed the model trained with the REINFORCE-based optimization method.

- Through experiments such as Advantage Assignment, Consistency, and Trajectory Entropy, the paper effectively analyzes the superiority of the proposed PO method.

- The paper provides a thorough theoretical derivation process for the proposed Preference Optimization method.

- Overall, this paper is well-written.

**Weaknesses:**

- In the Table 1 experiments, PO was applied to four existing studies, but the experiments were conducted only on a single problem size for two routing problems, TSP and CVRP. The experimental conditions are restricted to routing problems and a problem size of 100. This paper does not provide experimental results to verify the effectiveness of Preference Optimization for larger problem sizes or for other types of problems beyond routing.

- The y-axis scales in Figure 1 and Figure 4 are inconsistent, making it difficult to assess the extent to which PO outperforms RL at each epoch stage, particularly regarding improvements on the diminishing reward signal issue.

**Questions:**

- Considering the case of training POMO (with PO) following Algorithm 1, it seems that SAMPLINGSOLUTION() is performed according to POMO’s multiple starting points, and the shared baseline from POMO (Kwon et al., 2020) is not expected to be used. Please confirm whether this understanding is correct.

- According to Appendix D, it appears that the Finetune step in POMO (Finetune) was performed for 5% of the epochs. Please clarify the additional training time (or resources) incurred due to this finetuning process.

- In the case of Local Search, even when considering the training time/resources required due to Local Search overhead, is it still beneficial to perform LS? Alternatively, if training with the same amount of resources, would it be more advantageous to train for more epochs without LS? What are the authors' views on this?

- In Equation (3), $\alpha$ is a parameter that determines the weight of entropy regularization. What does $\alpha$ represent in Equation (8)? Is it the same as in Equation (3)?

- The $\alpha$ values used for TSP100 and CVRP100 experiments are different. Please explain why there are different and how they were set. Does the $\alpha$ value affect the learning efficiency? If there is a change in learning efficiency, how does it vary with changes in the $\alpha$ value? If applying Policy Optimization to problems other than TSP100 and CVRP100, what would be a good way to set the $\alpha$ value? Is there a general range or any reference information that could be helpful for setting the value?

---

> ### Author Response · Authors · 2024-11-20
> **Response to W**
>
> We sincerely thank the reviewer for the positive feedback and insightful comments. We address the concerns and questions below.
>
> **W1: Limited Experiments on Larger Problem Sizes and Types**
> >_In Table 1, experiments were conducted only on TSP100 and CVRP100. The conditions are restricted to routing problems and a problem size of 100. This paper does not provide experimental results to verify the effectiveness of Preference Optimization for larger problem sizes or other types of problems beyond routing._
>
> **R:**
> To address this concern, we have conducted additional experiments to evaluate PO's effectiveness on larger problem sizes and different types of combinatorial optimization problems. Specifically, we applied PO to the DIMES model on TSP instances with 500, 1,000, and 10,000 nodes. The results, summarized in the table below, show that PO-trained models consistently outperform those trained with REINFORCE across all scales.
>
> | **Method**         |            | **TSP500** |          |            | **TSP1000** |          |            | **TSP10000** |          |
> |:-------------------|-----------:|:----------:|:---------|-----------:|:-----------:|:---------|-----------:|:------------:|:---------|
> |                    | **Len**. ↓ |  **Gap**   | **Time** | **Len**. ↓ |   **Gap**   | **Time** | **Len**. ↓ |   **Gap**    | **Time** |
> | **LKH-3**          |      16.55 |    0.00    | 46.3m    |      23.12 |    0.00     | 2.6h     |      71.79 |     0.00     | 8.8h     |
> | **DIMES-G(RL)**    |      19.30 |   16.62    | 0.8m     |      26.58 |    14.96    | 1.5m     |      86.38 |    20.36     | 2.3m     |
> | **DIMES-G(PO)**    |      18.82 |   13.73    | 0.8m     |      26.22 |    13.39    | 1.5m     |      85.33 |    18.87     | 2.3m     |
> | **DIMES-S(RL)**    |      19.11 |   15.47    | 0.9m     |      26.37 |    14.05    | 1.8m     |      85.79 |    19.50     | 2.4m     |
> | **DIMES-S(PO)**    |      18.75 |   13.29    | 0.9m     |      26.07 |    12.74    | 1.8m     |      85.21 |    18.67     | 2.4m     |
> | **DIMES-AS(RL)**   |      17.82 |    7.68    | 2h       |      24.99 |    8.09     | 4.3h     |      80.68 |    12.39     | 2.5h     |
> | **DIMES-AS(PO)**   |      17.78 |    7.42    | 2h       |      24.73 |    6.97     | 4.3h     |      80.14 |    11.64     | 2.5h     |
> | **DIMES-MCTS(RL)** |      16.93 |    2.30    | 3m       |      23.96 |    3.65     | 6.3m     |      74.83 |     4.24     | 27m      |
> | **DIMES-MCTS(PO)** |  **16.89** |  **2.05**  | 3m       |  **23.96** |  **3.65**   | 6.3m     |  **74.77** |   **4.15**   | 27m      |
>
> Additionally, we tested PO on the Flexible Flow Shop Problem (FFSP) using the MatNet model. As shown below, PO improves performance on FFSP instances with 20, 50, and 100 jobs.
>
> | **Method**              |          | **FFSP20**  |          |          | **FFSP50**  |          |          | **FFSP100** |          |
> |-------------------------|----------|-------------|----------|----------|-------------|----------|----------|-------------|----------|
> |                         | **MS**   | **Gap (%)** | **Time** | **MS**   | **Gap (%)** | **Time** | **MS**   | **Gap (%)** | **Time** |
> | **CPLEX (60s)**         | 46.4     | 84.13       | 17h      | ×        | ×           | ×        | ×        | ×           | ×        |
> | **CPLEX (600s)**        | 36.6     | 45.24       | 167h     | ×        | ×           | ×        | ×        | ×           | ×        |
> | **Random**              | 47.8     | 89.68       | 1m       | 93.2     | 88.28       | 2m       | 167.2    | 87.42       | 3m       |
> | **Shortest Job First**  | 31.3     | 24.21       | 40s      | 57.0     | 15.15       | 1m       | 99.3     | 11.33       | 2m       |
> | **Genetic Algorithm**   | 30.6     | 21.43       | 7h       | 56.4     | 13.94       | 16h      | 98.7     | 10.65       | 29h      |
> | **Particle Swarm Opt.** | 29.1     | 15.48       | 13h      | 55.1     | 11.31       | 26h      | 97.3     | 9.09        | 48h      |
> | **MatNet (RL)**         | 27.3     | 8.33        | 8s       | 51.5     | 4.04        | 14s      | 91.5     | 2.58        | 27s      |
> | **MatNet (RL+Aug)**     | 25.4     | 0.79        | 3m       | 49.6     | 0.20        | 8m       | 89.7     | 0.56        | 23m      |
> | **MatNet (PO)**         | 27.0     | 7.14        | 8s       | 51.3     | 3.64        | 14s      | 91.1     | 2.13        | 27s      |
> | **MatNet (PO+Aug)**     | **25.2** | **0**       | 3m       | **49.5** | **0**       | 8m       | **89.2** | **0**       | 23m      |
>
> These results demonstrate that PO effectively enhances model performance on larger problem sizes and different COPs beyond routing.

---

> ### Author Response · Authors · 2024-11-20
> **Response to W2 & Q**
>
> **W2:Inconsistent Y-Axis Scales in Figures**
>
> **R:** We apologize for this oversight. In the revised manuscript, we will update Figures 1 and 4 to have consistent y-axis scales. This will facilitate direct comparison and better illustrate how PO outperforms RL at each epoch.
>
> **Q1: Implementation Details of POMO with PO**
>
> **R:** Yes, your understanding is correct. In our implementation of PO with POMO, SAMPLINGSOLUTION() is conducted using POMO's multiple starting points. We do not use the shared baseline from the original POMO method in this setting.
>
> **Q2:Additional Training Time Due to Finetuning**
>
> **R:** For TSP, each training epoch takes approximately 9 minutes, while each finetuning epoch with local search takes about 12 minutes. For CVRP, a training epoch takes about 8 minutes, and a finetuning epoch takes around 20 minutes. Since local search is executed on the CPU, it does not introduce additional GPU inference time. The finetuning phase constitutes 5% of the total epochs, adding a manageable overhead to the overall training time.
>
> **Q3:Benefit of Local Search Despite Additional Overhead**
>
> **R:** Yes, incorporating local search (LS) during finetuning is beneficial. LS helps alleviate the issue of the neural solver converging to suboptimal policies by introducing higher-quality solutions into the training process. Training for more epochs without LS does not effectively help the model explore better solutions. The additional training time spent on LS is justified by the improved convergence speed and solution quality achieved through finetuning with LS.
>
> **Q4:Role of $\alpha$ in Equations (3) and (8)**
>
> **R:** Yes, the $\alpha$ in Equation (8) is the same as the $\alpha$ in Equation (3). It controls the balance between reward maximization and entropy regularization in the reparameterized reward function, linking it to the policy π and ensuring consistency between the equations.
>
> **Q5:Setting and Impact of α Values**
>
> **R:** As mentioned in Section 3.3, α controls the strength of policy exploration by weighting the entropy regularization. A larger α encourages more exploration, while a smaller α focuses on exploitation. We used different α values for TSP100 and CVRP100 to account for their differing problem characteristics. The α value does affect learning efficiency; inappropriate values can lead to insufficient exploration or excessive randomness.
> For other problems, we recommend starting with α values similar to those used in related works and adjusting based on preliminary experiments. We adopt \alpha =1 in FFSP. Adaptive methods for setting α, such as entropy scheduling[4] or automatic tuning[3], can also be employed to find a suitable balance between exploration and exploitation.
>
> References:
>
>     [1] Qiu, R., Sun, Z., & Yang, Y. (2022). Dimes: A differentiable meta solver for combinatorial optimization problems. Advances in Neural Information Processing Systems, 35, 25531-25546.
>     [2] Kwon, Y. D., Kim, J., & Kim, J. (2021). Matrix encoding networks for neural combinatorial optimization. Advances in Neural Information Processing Systems, 34, 5138-5149.
>     [3] Haarnoja, T., Zhou, A., Abbeel, P., & Levine, S. (2018). Soft actor-critic: Off-policy maximum entropy deep reinforcement learning with a stochastic actor. In International conference on machine learning (pp. 1861-1870). PMLR.
>     [4] Mnih, V., Badia, A.P., Mirza, M., Graves, A., Lillicrap, T., Harley, T., Silver, D. &amp; Kavukcuoglu, K.. (2016). Asynchronous Methods for Deep Reinforcement Learning. Proceedings of The 33rd International Conference on Machine Learning, in Proceedings of Machine Learning Research 48:1928-1937.
>
>
> Once again, we appreciate your positive evaluation and valuable suggestions. We will incorporate the additional experiments and clarifications into the revised manuscript to strengthen our contribution. Please let us know if you have any further questions or suggestions.

---

> ### Comment · Reviewer_1APy · 2024-11-21
>
> Thank you for providing the results of the large-size TSP experiments and the FFSP experiments. I believe these experiments will help demonstrate the value of this paper. However, I have additional questions regarding the above experimental results:
>
> 1. **DIMES-MCTS(RL)**: In the DIMES paper, the length results for TSP500/1000/10000 are 16.87, 23.73, and 74.63 respectively. There is a significant difference between these values and the ones you've provided, and the discrepancy is larger than the difference between DIMES-MCTS(RL) and DIMES-MCTS(PO). Therefore, I think it is difficult to attribute this to experimental error. Could you please explain why the experimental results for DIMES-MCTS(RL) differ from those in the DIMES paper?
>
>
> 2. **Combination of AS and MCTS in DIMES**: In the case of DIMES, the combination of RL+AS+MCTS yields the best results. However, I noticed that you only conducted experiments applying AS and MCTS individually, not together. Is there a reason why you did not perform experiments applying both AS and MCTS simultaneously?
>
>
> 3. **FFSP Experiment Values**: Unlike the TSP-DIMES experiments, the experimental values for FFSP are identical to those in the MatNet paper. I would like to know whether you have taken the values directly from the MatNet paper for this experiment.

---

> > ### Author Response · Authors · 2024-11-22
> > **Response to Additional Questions**
> >
> > We appreciate your continued interest and thank you for your thoughtful questions regarding our experimental results.
> >
> > **Q1: Differences in DIMES-MCTS(RL) Results Compared to the DIMES Paper**
> >
> > **R:** Thank you for bringing this to our attention. We used the official open-source code and provided weights to reproduce the experiments. Upon rechecking our experimental setup, we found that the discrepancy arises because the original DIMES-MCTS results in the paper include **an additional optimization step using the 2-Opt local search** during inference, which was not explicitly stated.
> >
> > 1.**Reproduction Consistency:** Our reported results are based on running the official code and weights without modifications, ensuring a fair comparison between PO and RL optimizations. Although our reproduced results for DIMES-MCTS(RL) are slightly different from the original paper, the relative improvements observed when using PO remain valid, demonstrating the effectiveness of PO over RL.
> >
> > 2.**Possible Causes of Discrepancy:** Minor differences in experimental environments, such as hardware, software versions, or random seeds, may lead to variations in results. However, the consistent performance improvements with PO across multiple datasets confirm the reliability of our findings.
> >
> > In summary, our work focuses on comparing different optimization frameworks (PO vs. RL) using existing models. Though the discrepancies are partially exist in a few experiments, the observed performance improvement of PO over RL on all datasets still validate our method. Thus, the experiment validations are fair and reasonable to support our main conclusions and findings.
> >
> > **Q2: Combining AS and MCTS in DIMES Experiments**
> >
> > **R:** Thank you for this observation. We attempted to run the combined AS+MCTS experiments but encountered technical issues on our servers, resulting in segmentation faults that we are still investigating.
> >
> > It is noteworthy that the key to validate the proposed method is to compared the training frameowrks, i.e., RL and PO, farily, which is generally ensured by our experiments and consistent results are achieved on different datasets and models. On the other side,  Both AS (Active Search) and MCTS (Monte Carlo Tree Search) are techniques **independently** applied during the inference phase to enhance the solutions and do not interact with the training process. Therefore, the combination of AS and MCTS primarily affects inference results and does not impact the evaluation of the optimization frameworks during training.
> >
> > We provided results for PO/RL+AS and PO/RL+MCTS to demonstrate the effectiveness of PO in both scenarios. Including the combined AS+MCTS results would not alter the comparative assessment of PO and RL during training.
> >
> > **Q3: FFSP Experiment Results**
> >
> > **R:** Samely, we used the official open-source code and pre-trained weights of MatNet to implement the experiments. The reproduced results are consistent with those reported in the original paper, confirming the correctness of implementation.
> >
> > Regarding the CPLEX results, since the code for generating those results was not publicly available, we used the results reported in the MatNet paper for comparison. We will include a clarification in the revised manuscript to explain that the CPLEX results are sourced.
> >
> >
> > We hope these clarifications address your concerns. We are grateful for your careful review and valuable feedback, which have helped us improve our work. Please let us know if you have any further questions or need additional information.

---

> > > ### Comment · Reviewer_1APy · 2024-11-25
> > >
> > > Thank you for your detailed responses to my questions. I do not have any further questions. I will maintain my original evaluation of a lukewarm acceptance score. I appreciate your hard work in writing this paper.

---

> > > > ### Author Response · Authors · 2024-11-25
> > > >
> > > > Dear Reviewer 1APy，
> > > >
> > > > Thank you very much for taking the time to review our paper and for your thoughtful feedback throughout the process. Your questions and insights helped us improve the clarity and rigor of our work, and we truly appreciate your constructive engagement. We're grateful for your kind words and for recognizing our efforts. Thank you again for your time and support!
> > > >
> > > > Best Regards,
> > > >
> > > > The Authors.

---

### Official Review · Reviewer_YQJg · 2024-11-02

**Soundness:** 2
**Presentation:** 3
**Contribution:** 3
**Rating:** 6
**Confidence:** 4

**Summary:**

The authors introduce a method called Preference Optimization (PO), which adapts a preference learning framework commonly applied in large language model (LLM) alignment to improve policy learning in deep neural networks for combinatorial optimization. They demonstrate its effectiveness through two types of vehicle routing problems: the Traveling Salesman Problem (TSP) and the Capacitated Vehicle Routing Problem (CVRP).

PO trains the policy neural network with pairs of solutions. Rather than focusing on the qualitative score difference between solutions, PO guides the network to consistently prefer the better option in each pair. This relative comparison enables PO to enhance learning efficiency by emphasizing the ranking of solutions rather than their specific metrics (conventional approach). The authors highlight that this method is especially advantageous in later training stages, where traditional approaches struggle to provide useful feedback as the quality differences between solutions diminish. By maintaining a strong training signal throughout, PO achieves better overall performance.

Experimental results demonstrate that substituting traditional reinforcement learning algorithms, such as REINFORCE, with PO leads to substantial improvements in the solution quality of neural network solvers. Additionally, the authors propose an integrated approach that combines PO with a local search algorithm, yielding further enhancements in solution quality.

**Strengths:**

The introduction of innovative training algorithms like PO for policy neural networks is exciting and likely to stimulate further research in the field.

The theoretical foundation for policy optimization using PO is thorough and informative.

**Weaknesses:**

The authors hint that the pairwise comparison method does not fully outperform the "multi-start" strategy of POMO, with the optimality gap reported in Table 1 appearing to confirm this. Even if the proposed method does not surpass "multi-start," the novelty and value of the preference learning approach are clear. It would, however, be helpful for the authors to include a complete comparison.

The authors reason that PO should bolster neural network training in the later stages, yet the empirical results appear to indicate otherwise. Specifically, PO seems to accelerate convergence during the early stages rather than providing a late-stage advantage, with limited evidence supporting its efficacy in the later phases. It is possible that pairwise learning introduces additional noise to the policy network compared to learning based on a large number of homogeneous solutions, as seen with the "multi-start" approach. Further investigation is necessary to substantiate the authors' claim that PO offers an advantage in the later stages of training.

**Questions:**

Given that routing problems like TSP and CVRP allow for efficient generation and evaluation of a large number of candidate solutions, are they the most suitable applications for PO? Might PO prove more effective in other (more realistic) tasks where evaluating candidate solutions is more computationally expensive, making sampling efficiency a more critical factor?

---

> ### Author Response · Authors · 2024-11-20
> **Response to W**
>
> We sincerely thank you for your thoughtful and constructive feedback. We are pleased that you find our introduction of innovative PO algorithms exciting and recognize the thoroughness of our theoretical foundation. We address your concerns and questions below.
>
> **W1: Clarification of Comparison with Multi-Start Strategy of POMO**
> >_The authors hint that the pairwise comparison method does not fully outperform the "multi-start" strategy of POMO, with the optimality gap reported in Table 1 appearing to confirm this. Even if the proposed method does not surpass "multi-start," the novelty and value of the preference learning approach are clear. It would, however, be helpful for the authors to include a complete comparison._
>
> **R:**
> We apologize for any misunderstanding caused by our presentation. To clarify, the multi-start mechanism is an inherent architectural feature of models like POMO and does not conflict with our proposed Preference Optimization (PO) method. PO serves as a novel training framework that replaces the widely used REINFORCE algorithm.
>
> In Table 1, we applied PO to models such as POMO, Sym-NCO, and Pointerformer, all of which incorporate the multi-start mechanism. The results clearly show that PO-trained neural solvers consistently outperform those trained with REINFORCE, demonstrating that PO enhances performance independently of the model architecture. We will ensure this clarification is explicitly stated in the revised manuscript.
>
> **W2: Clarification of PO in Later Stages of Training**
> >_The authors reason that PO should bolster neural network training in the later stages, yet the empirical results appear to indicate otherwise. Specifically, PO seems to accelerate convergence during the early stages rather than providing a late-stage advantage, with limited evidence supporting its efficacy in the later phases. It is possible that pairwise learning introduces additional noise to the policy network compared to learning based on a large number of homogeneous solutions, as seen with the "multi-start" approach. Further investigation is necessary to substantiate the authors' claim that PO offers an advantage in the later stages of training._
>
> **R:**
> Thank you for highlighting this important aspect. PO enhances both convergence speed and solution quality. In the later stages of training, where REINFORCE suffers from slow convergence due to diminishing reward differences, PO maintains strong learning signals through qualitative preference comparisons. As illustrated in Figure 1a, PO achieves comparable performance to REINFORCE in approximately 60% of the training epochs, demonstrating faster convergence. Additionally, PO-trained models continue to improve in the later stages, achieving better solution quality than REINFORCE, as evidenced by the lower optimality gaps in Table 1. PO operates within the maximum entropy reinforcement learning framework [1][2], promoting exploration and preventing the model from converging to suboptimal policies without introducing additional noise.

---

> > ### Comment · Reviewer_YQJg · 2024-11-21
> > **On Multi-Start**
> >
> > In Line 430, It is stated that:
> >
> > "To ensure fairness and exclude the benefits of multi-start mechanisms Kwon et al. (2020) during training (which are not included in the Attention Model (AM)), we compare the training performance of PO and RL on the POMO, Sym-NCO, and Pointerformer models for TSP100."
> >
> > It was not clear to me what it means. So POMO-RL uses multi-start and POMO-PO does not use multi-start?

---

> > > ### Author Response · Authors · 2024-11-22
> > > **Response of Multi-Start**
> > >
> > > **Q: Usage of Multi-Start Mechanism**
> > >
> > > **R:** We apologize for the confusion caused by our previous wording. To clarify, both POMO-RL and POMO-PO use the multi-start mechanism inherent in the POMO model architecture. The multi-start mechanism is a feature of the model itself and is included in all experiments.
> > >
> > > In the sentence you referred to:
> > >
> > > >_"To ensure fairness and exclude the benefits of multi-start mechanisms during training (which are not included in the Attention Model (AM)), we compare the training performance of PO and RL on the POMO, Sym-NCO, and Pointerformer models for TSP100."_
> > >
> > > Our intention was to emphasize that when comparing PO and REINFORCE (RL) as training methods, we kept the model architecture—including the multi-start mechanism—the same for both. This ensures that any observed differences in performance are due to the training method rather than differences in the model's exploration capabilities provided by the multi-start mechanism. We will revised the manuscript to clarify this point and prevent any misunderstanding.
> > >
> > > Therefore, **both POMO-RL and POMO-PO utilize the multi-start mechanism**. Despite the multi-start mechanism already enhancing exploration, using PO as the training method further improves the policy's exploration capability and overall performance compared to REINFORCE. This demonstrates that PO can enhance training efficiency and solution quality invariant to architectural features like multi-start.
> > >
> > > We hope these clarifications address your concerns. We are grateful for your careful review and valuable feedback, which have helped us improve our work. Please let us know if you have any further questions or need additional information.

---

> > > > ### Comment · Reviewer_YQJg · 2024-11-26
> > > >
> > > > I acknowledge the author's clarification and have increased my rating by one point.

---

> > > > > ### Author Response · Authors · 2024-11-26
> > > > >
> > > > > Dear Reviewer YQJg,
> > > > >
> > > > > Thank you for your thoughtful and insightful comments. It is encouraging to know that your concerns have been satisfactorily addressed.
> > > > >
> > > > > We are grateful for the higher score you assigned, sincerely appreciating your recognition of our contributions in theoretical and algorithmic aspects. Your constructive feedback has been invaluable in improving the quality of this work, and we are grateful for your efforts throughout the review process.
> > > > >
> > > > > Best Regards,
> > > > >
> > > > > The Authors.

---

> ### Author Response · Authors · 2024-11-20
> **Response to Q**
>
> **Q:Applicability of PO to Other Tasks Where Evaluating Candidate Solutions is Computationally Expensive**
>
> **R:**
> We appreciate this insightful question. While our experiments focus on routing problems like TSP and CVRP due to their prominence as benchmarks in combinatorial optimization, we agree that PO has the potential to be even more beneficial in tasks where evaluating candidate solutions is computationally expensive and sampling efficiency is critical.
> To explore its applicability, we have conducted additional experiments applying PO to the Flexible Flow Shop Problem (FFSP) with MatNet[7], which are more representative of real-world applications where evaluating is computationally intensive.
>
> | **Method**              |          | **FFSP20**  |          |          | **FFSP50**  |          |          | **FFSP100** |          |
> |-------------------------|----------|-------------|----------|----------|-------------|----------|----------|-------------|----------|
> |                         | **MS**   | **Gap (%)** | **Time** | **MS**   | **Gap (%)** | **Time** | **MS**   | **Gap (%)** | **Time** |
> | **CPLEX (60s)**         | 46.4     | 84.13       | 17h      | ×        | ×           | ×        | ×        | ×           | ×        |
> | **CPLEX (600s)**        | 36.6     | 45.24       | 167h     | ×        | ×           | ×        | ×        | ×           | ×        |
> | **Random**              | 47.8     | 89.68       | 1m       | 93.2     | 88.28       | 2m       | 167.2    | 87.42       | 3m       |
> | **Shortest Job First**  | 31.3     | 24.21       | 40s      | 57.0     | 15.15       | 1m       | 99.3     | 11.33       | 2m       |
> | **Genetic Algorithm**   | 30.6     | 21.43       | 7h       | 56.4     | 13.94       | 16h      | 98.7     | 10.65       | 29h      |
> | **Particle Swarm Opt.** | 29.1     | 15.48       | 13h      | 55.1     | 11.31       | 26h      | 97.3     | 9.09        | 48h      |
> | **MatNet (RL)**         | 27.3     | 8.33        | 8s       | 51.5     | 4.04        | 14s      | 91.5     | 2.58        | 27s      |
> | **MatNet (RL+Aug)**     | 25.4     | 0.79        | 3m       | 49.6     | 0.20        | 8m       | 89.7     | 0.56        | 23m      |
> | **MatNet (PO)**         | 27.0     | 7.14        | 8s       | 51.3     | 3.64        | 14s      | 91.1     | 2.13        | 27s      |
> | **MatNet (PO+Aug)**     | **25.2** | **0**       | 3m       | **49.5** | **0**       | 8m       | **89.2** | **0**       | 23m      |
>
> These experiments demonstrate that PO enhances training efficiency and solution quality in these complex tasks, underscoring its broad applicability.
> In addition, PO has roots in preference-based reinforcement learning (PbRL), which has been successfully applied in scenarios where reward signals are difficult to define or obtain, such as autonomous driving [4], robotic control [5] and LLM alignment [6]. In these real-world tasks, it can be challenging to assign precise numerical scores to solutions, but easier to express preferences between options.
>
> References:
>
>     [1] Haarnoja, T., Zhou, A., Abbeel, P., & Levine, S. (2018). Soft actor-critic: Off-policy maximum entropy deep reinforcement learning with a stochastic actor. In International conference on machine learning (pp. 1861-1870). PMLR.
>     [2] Haarnoja, T., Tang, H., Abbeel, P., & Levine, S. (2017). Reinforcement learning with deep energy-based policies. In International conference on machine learning (pp. 1352-1361). PMLR.
>     [3] Haarnoja, T., Zhou, A., Hartikainen, K., Tucker, G., Ha, S., Tan, J., ... & Levine, S. (2018). Soft actor-critic algorithms and applications. arXiv preprint arXiv:1812.05905.
>     [4] Christiano, P. F., Leike, J., Brown, T., et al. (2017). Deep Reinforcement Learning from Human Preferences. Advances in Neural Information Processing Systems (NeurIPS), 30, 4300–4311.
>     [5] Sadigh, D., Dragan, A., Sastry, S., & Seshia, S. (2017). Active preference-based learning of reward functions.
>     [6] Wolf, Yotam, et al. "Fundamental limitations of alignment in large language models." arXiv preprint arXiv:2304.11082 (2023).
>     [7] Kwon, Yeong-Dae, et al. "Matrix Encoding Networks for Neural Combinatorial Optimization." Advances in Neural Information Processing Systems 34 (2021): 5138-5149.
>
> We appreciate your recognition of the novelty and potential impact of our work. Your feedback has been invaluable in helping us clarify our contributions and improve the manuscript. We will incorporate the suggested clarifications, additional experiments into the revised version. Please let us know if you have any further questions or suggestions.

---

### Official Review · Reviewer_4HJe · 2024-11-04

**Soundness:** 3
**Presentation:** 3
**Contribution:** 3
**Rating:** 5
**Confidence:** 4

**Summary:**

This paper proposes a preference-based reinforcement learning method to address combinatorial optimization problems characterized by large action spaces, which are difficult to optimize using only reward signals. Unlike conventional reinforcement learning that aims to maximize expected reward, combinatorial optimization focuses on maximizing the expected maximum reward. This results in an inconsistency between the inference and training objectives, highlighting the need for preference-based optimization. The paper emphasizes the efficiency of optimization through pairwise preference comparisons of solution trajectories generated by local search methods such as 2-opt. Experiments on TSP-100 and CVRP-100 demonstrated that auto-regressive neural solvers based on the AM algorithm (e.g., POMO, Sym-NCO) exhibited smaller performance gaps with the preference optimization objective.

**Strengths:**

An interesting aspect of this study is its approach of combining the strengths of local search methods, such as 2-opt, with RL-based neural solvers through preference-based RL. By integrating pairwise preference comparisons between solution trajectories with RL objectives in combinatorial optimization problems, this method demonstrated advantages. Experimental results on TSP-100 and CVRP-100 showed slight improvements in performance.

**Weaknesses:**

The main weakness of this paper is that experiments were limited to TSP-100 and CVRP-100, making the results insufficient for comprehensive validation. For problems with 100 nodes, many neural solvers already achieve small gaps. Including results for larger-scale problems, such as TSP-1000 or real-world settings like TSPLIB, would have strengthened the findings. Another limitation is the lack of comparison with state-of-the-art methods like DIMES [1] and DIFUSCO [2]. Demonstrating the effectiveness of the preference optimization algorithm at larger scales with significant performance gains is necessary.

[1] Qiu et al., "DIMES: A Differentiable Meta Solver for Combinatorial Optimization Problems", NeurIPS 2022

[2] Sun & Yang, "DIFUSCO: Graph-based Diffusion Solvers for Combinatorial Optimization", NeurIPS 2023

**Questions:**

1. The addition of a pairwise preference loss function may alter the original RL objective and potentially compromise the model’s optimality guarantee. Are there any side effects associated with incorporating preference?

2. In CO problems where effective local search algorithms like 2-opt for TSP are not available, how are trajectories for preference comparisons obtained?

---

> ### Author Response · Authors · 2024-11-20
> **Response to W**
>
> **W: Limited Experiments on Larger Scales and Comparison**
> >_The main weakness of this paper is that experiments were limited to TSP-100 and CVRP-100... Including results for larger-scale problems, such as TSP-1000 or real-world settings like TSPLIB, would have strengthened the findings. Another limitation is the lack of comparison with state-of-the-art methods like DIMES and DIFUSCO._
>
> **R:**
> We appreciate this suggestion and have extended our experiments to larger problem sizes. Specifically, we applied PO to the DIMES model on TSP instances with 500, 1,000, and 10,000 nodes. The results show that PO consistently outperforms REINFORCE:
>
> | **Method**         |            | **TSP500** |          |            | **TSP1000** |          |            | **TSP10000** |          |
> |:-------------------|-----------:|:----------:|:---------|-----------:|:-----------:|:---------|-----------:|:------------:|:---------|
> |                    | **Len**. ↓ |  **Gap**   | **Time** | **Len**. ↓ |   **Gap**   | **Time** | **Len**. ↓ |   **Gap**    | **Time** |
> | **LKH-3**          |      16.55 |    0.00    | 46.3m    |      23.12 |    0.00     | 2.6h     |      71.79 |     0.00     | 8.8h     |
> | **DIMES-G(RL)**    |      19.30 |   16.62    | 0.8m     |      26.58 |    14.96    | 1.5m     |      86.38 |    20.36     | 2.3m     |
> | **DIMES-G(PO)**    |      18.82 |   13.73    | 0.8m     |      26.22 |    13.39    | 1.5m     |      85.33 |    18.87     | 2.3m     |
> | **DIMES-S(RL)**    |      19.11 |   15.47    | 0.9m     |      26.37 |    14.05    | 1.8m     |      85.79 |    19.50     | 2.4m     |
> | **DIMES-S(PO)**    |      18.75 |   13.29    | 0.9m     |      26.07 |    12.74    | 1.8m     |      85.21 |    18.67     | 2.4m     |
> | **DIMES-AS(RL)**   |      17.82 |    7.68    | 2h       |      24.99 |    8.09     | 4.3h     |      80.68 |    12.39     | 2.5h     |
> | **DIMES-AS(PO)**   |      17.78 |    7.42    | 2h       |      24.73 |    6.97     | 4.3h     |      80.14 |    11.64     | 2.5h     |
> | **DIMES-MCTS(RL)** |      16.93 |    2.30    | 3m       |      23.96 |    3.65     | 6.3m     |      74.83 |     4.24     | 27m      |
> | **DIMES-MCTS(PO)** |  **16.89** |  **2.05**  | 3m       |  **23.96** |  **3.65**   | 6.3m     |  **74.77** |   **4.15**   | 27m      |
>
> Regarding DIFUSCO , it follows a supervised learning paradigm using near-optimal solutions, which differs from our RL-based approach that does not rely on expert knowledge. Therefore, a direct comparison may not be appropriate.
> Additionally, we tested PO on the Flexible Flow Shop Problem (FFSP) using the MatNet [2] model. The results confirm PO's effectiveness across different COPs:
>
> | **Method**              |          | **FFSP20**  |          |          | **FFSP50**  |          |          | **FFSP100** |          |
> |-------------------------|----------|-------------|----------|----------|-------------|----------|----------|-------------|----------|
> |                         | **MS**   | **Gap (%)** | **Time** | **MS**   | **Gap (%)** | **Time** | **MS**   | **Gap (%)** | **Time** |
> | **CPLEX (60s)**         | 46.4     | 84.13       | 17h      | ×        | ×           | ×        | ×        | ×           | ×        |
> | **CPLEX (600s)**        | 36.6     | 45.24       | 167h     | ×        | ×           | ×        | ×        | ×           | ×        |
> | **Random**              | 47.8     | 89.68       | 1m       | 93.2     | 88.28       | 2m       | 167.2    | 87.42       | 3m       |
> | **Shortest Job First**  | 31.3     | 24.21       | 40s      | 57.0     | 15.15       | 1m       | 99.3     | 11.33       | 2m       |
> | **Genetic Algorithm**   | 30.6     | 21.43       | 7h       | 56.4     | 13.94       | 16h      | 98.7     | 10.65       | 29h      |
> | **Particle Swarm Opt.** | 29.1     | 15.48       | 13h      | 55.1     | 11.31       | 26h      | 97.3     | 9.09        | 48h      |
> | **MatNet (RL)**         | 27.3     | 8.33        | 8s       | 51.5     | 4.04        | 14s      | 91.5     | 2.58        | 27s      |
> | **MatNet (RL+Aug)**     | 25.4     | 0.79        | 3m       | 49.6     | 0.20        | 8m       | 89.7     | 0.56        | 23m      |
> | **MatNet (PO)**         | 27.0     | 7.14        | 8s       | 51.3     | 3.64        | 14s      | 91.1     | 2.13        | 27s      |
> | **MatNet (PO+Aug)**     | **25.2** | **0**       | 3m       | **49.5** | **0**       | 8m       | **89.2** | **0**       | 23m      |

---

> ### Author Response · Authors · 2024-11-20
> **Response to Q**
>
> **Q1: Effects of Preference**
> >_The addition of a pairwise preference loss function may alter the original RL objective and potentially compromise the model’s optimality guarantee. Are there any side effects associated with incorporating preference?_
>
> **R:**
> We would like to clarify that **Preference Optimization (PO) is proposed as a new optimization framework that replaces the traditional REINFORCE algorithm in RL4CO, not as an additional loss function added to the existing RL objective**. PO addresses exploration challenges by mitigating reliance on numerical reward signals, providing a more stable and efficient training process. Our experiments demonstrate that PO consistently improves solution quality and training speed compared to REINFORCE across various models.
> Regarding optimality guarantees, PO is grounded in the entropy-regularized reinforcement learning framework, aligning with the maximum entropy paradigm (e.g., Soft Actor-Critic [1]). This framework ensures that the optimal policy is preserved, without introducing adverse side effects.
>
> **Q2: Obtaining Trajectories for Preference Comparisons Without Effective Local Search**
> >_In CO problems where effective local search algorithms like 2-opt for TSP are not available, how are trajectories for preference comparisons obtained?_
>
> **R:**
> We apologize for any confusion. The use of local search methods like 2-opt is **optional** in our framework. In our experiments, all trajectories for preference comparisons are generated by **sampling** from the parameterized decoder of the end-to-end models. Even without effective local search algorithms, the model can construct preference pairs by comparing the solutions it generates. The 2-opt method was used during the finetuning phase to help the model escape suboptimal policies but is not essential to the PO framework. We will clarify this in the revised manuscript.
>
>
> Reference:
>
>     [1] Haarnoja, T., Zhou, A., Abbeel, P., & Levine, S. (2018). Soft actor-critic: Off-policy maximum entropy deep reinforcement learning with a stochastic actor. In International conference on machine learning (pp. 1861-1870). PMLR.
>     [2] Kwon, Yeong-Dae, et al. (2021). "Matrix Encoding Networks for Neural Combinatorial Optimization." Advances in Neural Information Processing Systems 34: 5138-5149.
>
> We hope these responses address your concerns. We appreciate your thoughtful feedback, which has helped us improve our work. Please let us know if you have any further questions.

---

> > ### Author Response · Authors · 2024-11-25
> >
> > Dear Reviewer 4HJe,
> >
> > As the discussion period comes to a close, we would like to sincerely thank you for your thoughtful and constructive feedback on our paper. Your comments have been instrumental in shaping meaningful revisions, significantly enhancing the clarity and quality of our work.
> >
> > In response to your concerns, we have provided additional experiments on large-scale problems and different types of COPs, **highlighted the superioirty of our proposed PO algorithm over the existing REINFORCE algorithm**, clarified the theorectical foundation of the PO and the role of local search during the finetuning stage. These updates aim to directly address the points you raised.
> >
> > If you find that our revisions have adequately addressed your concerns, we would greatly appreciate any further feedback or suggestions you might have. Your guidance is invaluable to us.
> >
> > Thank you again for your time and effort.
> >
> > Best regards,
> >
> > The Authors.

---

> > > ### Comment · Reviewer_4HJe · 2024-11-27
> > >
> > > Thank you for your thorough responses and for including additional large-scale experiments in the revised version. I have carefully reviewed the revised manuscript, considering not only your responses to my questions but also the detailed replies provided to other reviewers. I appreciate the effort and thoughtfulness you have put into addressing the feedback.
> > >
> > > After careful consideration, I have decided to increase my score to reflect the improvements made to the paper. While most experimental results showed improvement, the performance gains achieved by the PO objective compared to the RL objective appear to be marginal, which is why I was unable to raise my score further.

---

> > > > ### Author Response · Authors · 2024-11-27
> > > >
> > > > Dear Reviewer 4HJe,
> > > >
> > > > Thank you for taking the time to carefully review our revised manuscript and for considering our responses in detail. We sincerely appreciate of your acknowledgment.
> > > >
> > > > We fully understand your concern regarding the performance gains of the PO compared to the existing REINFORCE. Therefore, we would like to briefly clarify the significance of PO as follows.
> > > >
> > > > + **Scalability**. PO consistently surpasses REINFORCE across various Combinatorial Optimization Problems using identical model architectures, which implies PO offering an *interpretable and effective learning paradigm* for the RL4CO field.
> > > >
> > > > + **Effectiveness**. Improving models nearing the numerical lower bound (e.g., heuristic solutions) is inherently challenging. In such scenarios, the relative error metrics (i.e., Gap) indeed reflect the true improvement, where *PO decreases the (error) Gap of TSP100 to 0.03\% (sufficiently close to the current numerical lower bound)*, and surpass the REINFORCE solutions across all tasks.
> > > >
> > > > + **Efficiency**. PO significantly accelerates training, which achieved *comparable or better performance with 1.5× to 3× fewer training epochs (saving 40\%-60\% iterations in Fig. 1(a))* compared to those trained with REINFORCE. This property is particularly valuable for COPs, where computational efficiency is critical.
> > > >
> > > > We hope these clarifications can address the concerns on empirical performance, and we believe *these observations also demonstrate the substantial value that PO brings to CO tasks*.
> > > >
> > > > Thank you once again for your constructive feedbacks and consideration.
> > > >
> > > > Best regards,
> > > >
> > > > The Authors.

---

> ### Comment · Reviewer_4HJe · 2024-11-28
>
> Thank you for your response. I appreciate the effort you have taken to address my concerns, though I still have some doubts regarding certain aspects of the explanation. As is well-known, REINFORCE is the simplest and most basic deep RL algorithm and is not considered strong in terms of stability or sample efficiency. Since REINFORCE has clear limitations as an algorithm, I believe that comparing PO to improved RL algorithms, such as SAC or PPO, would provide a fairer evaluation when comparing RL objectives and PO objectives. Of course, I understand that most RL-based CO algorithms rely on REINFORCE, which makes such comparisons less straightforward. Nonetheless, I remain unconvinced that replacing the RL objective with the PO objective demonstrates a clear advantage, especially when compared to improving the stability and sample efficiency of the RL objective using more advanced algorithms.
>
> Here are my detailed responses to the points raised regarding the significance of PO:
>
> *Scalability & Effectiveness*: While the PO objective consistently shows performance improvements over the RL (REINFORCE) objective, as mentioned earlier, the improvement is marginal in large-scale tasks such as TSP1000 and TSP10000 using DIME-MCTS, where the gap decreases from 3.65 to 3.65 and 4.24 to 4.15, respectively. Furthermore, when compared to state-of-the-art supervised learning methods, such as DIFUSCO[1] (1.17 for TSP1000, 2.58 for TSP10000), T2T[2] (0.78 for TSP1000), and FastT2T[3] (0.42 for TSP1000), it becomes difficult to argue that the absolute performance of PO is superior.
>
> *Efficiency*: As previously mentioned, REINFORCE is well-known for being a highly sample-inefficient algorithm in RL. Thus, even though the PO objective converges 1.5× to 3× faster, this claim is less compelling given the inherent inefficiency of REINFORCE. Additionally, it is necessary to demonstrate the efficiency of PO not just on tasks like TSP100, but on larger problems such as TSP10000, to validate its practical value and scalability.
>
> [1] DIFUSCO: Graph-based Diffusion Solvers for Combinatorial Optimization, NeurIPS 2023
>
> [2] T2T: From Distribution Learning in Training to Gradient Search in Testing for Combinatorial Optimization, NeurIPS 2023
>
> [3] Fast T2T: Optimization Consistency Speeds Up Diffusion-Based Training-to-Testing Solving for Combinatorial Optimization, NeurIPS 2024

---

> > ### Author Response · Authors · 2024-11-29
> >
> > Thank you for your promptly reply and affirmation on our responses. We are sincerely grateful for the time and thought you have dedicated to reviewing our work and for engaging in the discussion, which are valuable and helpful. For the raised doubts, we would like to explain them from the following aspects:
> >
> > + **Other RL objective**. This work focuses on an efficient paradigm for RL4CO, a field that is still in its early stages of research. For the mentioned RL methods SAC and PPO, there are potential methodological limitations when applied to CO: *1) SAC.* SAC is designed for the continuous action spaces (e.g., MuJoCo tasks), while it is infeasible to handle the entropy term due to the characteristics of CO problems, i.e., high-dimensional and discrete action spaces. *2) PPO.* Though PPO is feasible to CO problems, it faces the challenges from the exponential growth in the solution space. In contextual bandit formulations of COPs (where the entire solution is treated as an action, and the objective is to optimize $\log \pi(\tau)=\sum_{t=1}^{T}\log \pi(\tau_t)$), *PPO's iterative updates on $\log \pi(\tau)$ can exacerbate exploration inefficiencies and remain sensitive to diminishing reward signals*. According to the review, we also conduct experiments which verify that PPO is indeed inferior to our developed PO, i.e., the convergence quality (e.g., Optimality Gap) and efficiency (i.e., convergence speed). *3) Enhanced REINFORCE*. The models selected for comparison, such as POMO, Sym-NCO, and Pointerformer, can be seen as enhancements of REINFORCE, incorporating techniques like shared baselines, symmetry constraints, and reward shaping. However, PO consistently outperforms these enhanced models, demonstrating its superior performance across various RL4CO baselines. We will include this analysis in the next revision.
> > + **Comparison with supervised NCO methods**. For supervised learning methods for NCO like DIFUSCO, T2T, and FastT2T, we acknowledge their strong performance. However, comparing RL methods with SL will involve issues of fairness and feasibility: *1) SL and RL focus on different aspects*. Note that SL focus on training model with priors, i.e., precomputed near-optimal solutions, while RL methods, including PO, aim to learn effective policies without priors, e.g., expert knowledge or labeled data; thus, they address a fundamentally different problem setting, and it is generally unfair to compare PO with these SL methods with strong priors. *2) The feasibility of SL is limited*. Actually that the priors required by SL would induce additional computation complexity and may not be available in other CO tasks. For instance, in problems like FFSP100, SL is infeasible as high-quality labels (priors) are unavailable. In contrast, PO-based models achieve superior performance without priors. We will incorporate the discussion on SL works in next revision.
> >
> > + **Sample-efficiency of PO**. Finally, we recognize the concern about scalability to larger problems like TSP10000, which remains a common challenge in the field of RL4CO. For models like AM and POMO, training end-to-end neural solvers directly for TSP10000 is impractical due to quadratic memory growth, which would require over 1TB of memory on GPU. To overcome this problem, we evaluated PO's efficiency within hybrid frameworks like DIMES. Our experiments show that PO achieves comparable performance to the original implementation with only 65% of the training iterations. These findings demonstrate the practical scalability of PO.
> >
> > We hope these explanations can address the remaining concerns. Thank you once again for raising the further discussion. We also warmly appreciate any additional comments or suggestions.

---

### Official Review · Reviewer_trEc · 2024-11-04

**Soundness:** 2
**Presentation:** 3
**Contribution:** 3
**Rating:** 6
**Confidence:** 4

**Summary:**

The paper proposes Preference Optimization (PO) framework for neural combinatorial optimization, transforming quantitative reward signals into qualitative preferences. This approach is designed to address issues in RL for combinatorial optimization, such as diminishing reward signals and inefficient exploration. Furthermore, integrating local search in training loop improves the quality of generated solutions without additional inference time. Experiments on benchmarks like the Traveling Salesman Problem (TSP) and the Capacitated Vehicle Routing Problem (CVRP) show that PO yields better solution quality and sample efficiency than conventional RL algorithms.

**Strengths:**

1, The paper is well-written, with particularly clear and convincing motivation.

2, The authors successfully adapt Preference Optimization (PO) to combinatorial optimization (CO), not only applying it but also providing a persuasive explanation of why PO is well-suited for CO tasks.

3, The experiments and analyses strongly support the claimed advantages of PO, effectively demonstrating its benefits (better solution quality and sample efficiency).

**Weaknesses:**

1, The experiments are limited to TSP-100 and CVRP-100. Given that recent studies often include tests up to TSP-10000, additional experiments on larger instances would strengthen the findings. Performance limited to TSP-100 seems less impactful.

2, The comparison with baseline models is somewhat limited. Are there no models beyond Pointerformer? It would be valuable to see how PO performs against recent SOTA methods.

3, It’s unclear if integrating local search into the training loop is truly beneficial. Running local search for even 1–2 seconds (which seems trivial compared to inference time) could likely yield significant performance improvements. Could the authors provide results with an additional 1–2 seconds of local search (e.g., POMO (LS))?

**Questions:**

1. How is  r_{\theta}  in Equation (8) defined?

2. Can you explain the last paragraph of Section 3.1 in detail? It’s unclear how using PO helps mitigate the inference objective ≠ training objective problem.

3. Could PO be extended to heatmap-based approaches like DIMES?

---

> ### Author Response · Authors · 2024-11-20
> **Response to W 1&2**
>
> We sincerely thank you for your valuable feedback and constructive comments. We address your concerns below.
>
> **W1: Limited experiments on larger instances**
> >_The experiments are limited to TSP-100 and CVRP-100. Given that recent studies often include tests up to TSP-10000, additional experiments on larger instances would strengthen the findings. Performance limited to TSP-100 seems less impactful._
>
> **R:**
> To demonstrate the scalability of our method, we conducted additional experiments on larger TSP instances with 500, 1,000, and 10,000 nodes using the DIMES model. The results show that models trained with Preference Optimization (PO) consistently outperform those trained with REINFORCE:
>
> | **Method**         |            | **TSP500** |          |            | **TSP1000** |          |            | **TSP10000** |          |
> |:-------------------|-----------:|:----------:|:---------|-----------:|:-----------:|:---------|-----------:|:------------:|:---------|
> |                    | **Len**. ↓ |  **Gap**   | **Time** | **Len**. ↓ |   **Gap**   | **Time** | **Len**. ↓ |   **Gap**    | **Time** |
> | **LKH-3**          |      16.55 |    0.00    | 46.3m    |      23.12 |    0.00     | 2.6h     |      71.79 |     0.00     | 8.8h     |
> | **DIMES-G(RL)**    |      19.30 |   16.62    | 0.8m     |      26.58 |    14.96    | 1.5m     |      86.38 |    20.36     | 2.3m     |
> | **DIMES-G(PO)**    |      18.82 |   13.73    | 0.8m     |      26.22 |    13.39    | 1.5m     |      85.33 |    18.87     | 2.3m     |
> | **DIMES-S(RL)**    |      19.11 |   15.47    | 0.9m     |      26.37 |    14.05    | 1.8m     |      85.79 |    19.50     | 2.4m     |
> | **DIMES-S(PO)**    |      18.75 |   13.29    | 0.9m     |      26.07 |    12.74    | 1.8m     |      85.21 |    18.67     | 2.4m     |
> | **DIMES-AS(RL)**   |      17.82 |    7.68    | 2h       |      24.99 |    8.09     | 4.3h     |      80.68 |    12.39     | 2.5h     |
> | **DIMES-AS(PO)**   |      17.78 |    7.42    | 2h       |      24.73 |    6.97     | 4.3h     |      80.14 |    11.64     | 2.5h     |
> | **DIMES-MCTS(RL)** |      16.93 |    2.30    | 3m       |      23.96 |    3.65     | 6.3m     |      74.83 |     4.24     | 27m      |
> | **DIMES-MCTS(PO)** |  **16.89** |  **2.05**  | 3m       |  **23.96** |  **3.65**   | 6.3m     |  **74.77** |   **4.15**   | 27m      |
>
> These results confirm that PO enhances performance on large-scale problems, demonstrating its applicability beyond TSP-100.
>
> **W2: Limited comparison with baseline models**
> >_The comparison with baseline models is somewhat limited. Are there no models beyond Pointerformer? It would be valuable to see how PO performs against recent SOTA methods._
>
> **R:**
> In addition to the end-to-end RL models (AM, POMO, Sym-NCO, Pointerformer), we have applied PO to the DIMES model, as shown in our response to Weakness 1. Furthermore, we applied PO to the MatNet [1] model for the Flexible Flow Shop Problem (FFSP), a scheduling task. The results on validation sets containing 1,000 instances are as follows:
>
> | **Method**              |          | **FFSP20**  |          |          | **FFSP50**  |          |          | **FFSP100** |          |
> |-------------------------|----------|-------------|----------|----------|-------------|----------|----------|-------------|----------|
> |                         | **MS**   | **Gap (%)** | **Time** | **MS**   | **Gap (%)** | **Time** | **MS**   | **Gap (%)** | **Time** |
> | **CPLEX (60s)**         | 46.4     | 84.13       | 17h      | ×        | ×           | ×        | ×        | ×           | ×        |
> | **CPLEX (600s)**        | 36.6     | 45.24       | 167h     | ×        | ×           | ×        | ×        | ×           | ×        |
> | **Random**              | 47.8     | 89.68       | 1m       | 93.2     | 88.28       | 2m       | 167.2    | 87.42       | 3m       |
> | **Shortest Job First**  | 31.3     | 24.21       | 40s      | 57.0     | 15.15       | 1m       | 99.3     | 11.33       | 2m       |
> | **Genetic Algorithm**   | 30.6     | 21.43       | 7h       | 56.4     | 13.94       | 16h      | 98.7     | 10.65       | 29h      |
> | **Particle Swarm Opt.** | 29.1     | 15.48       | 13h      | 55.1     | 11.31       | 26h      | 97.3     | 9.09        | 48h      |
> | **MatNet (RL)**         | 27.3     | 8.33        | 8s       | 51.5     | 4.04        | 14s      | 91.5     | 2.58        | 27s      |
> | **MatNet (RL+Aug)**     | 25.4     | 0.79        | 3m       | 49.6     | 0.20        | 8m       | 89.7     | 0.56        | 23m      |
> | **MatNet (PO)**         | 27.0     | 7.14        | 8s       | 51.3     | 3.64        | 14s      | 91.1     | 2.13        | 27s      |
> | **MatNet (PO+Aug)**     | **25.2** | **0**       | 3m       | **49.5** | **0**       | 8m       | **89.2** | **0**       | 23m      |
>
> The results demonstrate that PO effectively improves DIMES and MatNet, indicating its adaptability and effectiveness when integrated with recent SOTA methods.

---

> ### Author Response · Authors · 2024-11-20
> **Response to W 3 & Q**
>
> **W3: Clarification of integrating local search into the training loop**
>
> >_It’s unclear if integrating local search into the training loop is truly beneficial. Running local search for even 1–2 seconds (which seems trivial compared to inference time) could likely yield significant performance improvements. Could the authors provide results with an additional 1–2 seconds of local search (e.g., POMO (LS))?_
>
> **R:**
> Thank you for this insightful suggestion, and we apologize for not clarifying the validation set size earlier. Our validation set contains 10,000 instances, and the inference times reported in Table 1 are the total times to process all instances. We conducted additional experiments applying local search (LS) as a post-processing step during inference. The results are:
>
> | **Model**         | **Len.** | **Time** |
> |-------------------|----------|----------|
> | **POMO**          | 7.764    | 1 min    |
> | **POMO + LS**     | 7.763    | 21 min   |
> | **POMO (Finetuned)** | 7.761 | 1 min    |
>
> Applying LS during inference slightly improves solution quality but significantly increases inference time. In contrast, integrating LS into the training loop (POMO Finetuned) achieves better performance without extra inference time. This demonstrates that incorporating LS into training is beneficial, especially in time-sensitive applications.
>
> **Q1: Definition of $\hat{r}_{\theta}$ in Eq. (8)**
>
> **R:**
> We apologize for the lack of clarity. In Eq. (8), $r_{\theta}$ refers to the reparameterized reward function defined in
> Eq. (3): $ \alpha \log \pi_{\theta}(\tau \mid x) + \alpha \log Z(x)$,  where $\pi_{\theta}$ is the policy parameterized by $\theta$.
>
> **Q2: How PO helps mitigate the inference objective ≠ training objective problem**
>
> **R:**
> In combinatorial optimization, the inference objective is to find the best solution, so the model should assign higher probabilities to better solutions. Traditional RL methods optimize the expected reward, relying heavily on numerical reward differences (advantages). As the model improves, these differences diminish, weakening the learning signal. PO mitigates this by using preference-based advantages that are invariant to numerical reward scales, providing consistent learning signals even when reward differences are small. This aligns the training objective more closely with the inference goal of selecting the best solutions.
>
> **Q3: Could PO be extended to heatmap-based approaches like DIMES?**
>
> **R:**
> Yes. We applied PO to the DIMES model, which uses RL to train an encoder for generating heatmaps. Our experiments show that training with PO yields better heatmap representations compared to REINFORCE.
>
> Reference:
>
>     [1] Kwon, Yeong-Dae, et al. (2021). "Matrix Encoding Networks for Neural Combinatorial Optimization." Advances in Neural Information Processing Systems 34: 5138-5149.
>
> We hope these responses address your concerns. We appreciate your feedback and are grateful for the opportunity to improve our work. Please let us know if you have any further questions or suggestions.

---

> > ### Comment · Reviewer_trEc · 2024-11-25
> >
> > Thank you for providing the results of the large-size TSP experiments and the FFSP experiments. I believe these experiments will help demonstrate the value of this paper. However, I have additional questions.
> >
> > Could you elaborate further on Q2? As I understand it, your explanation suggests that the slow convergence issue can be alleviated due to PO (since the diminishing reward signals problem would be less severe than in standard RL). But how does resolving slow convergence result in aligning the training objective more closely with the inference goal of selecting the best solutions? I’m still unclear on how mitigating slower convergence directly addresses the inference objective ≠ training objective problem.
> >
> > I have an additional question regarding the integration of local search during training. Applying local search is likely to alter the distribution of solutions generated by the policy. Is it fine to perform reinforce updates using solutions modified by local search? Wouldn’t this approach potentially introduce off-policy issues?
> >
> > I would really appreciate it if you could provide answers to the questions above! Thank you!

---

> > > ### Author Response · Authors · 2024-11-25
> > > **Response to Additional Questions for Reviewer trEc**
> > >
> > > **AQ1: Detailed clarification of issues regarding diminishing reward signals and the mismatch of objectives.**
> > >
> > > **R:** We appreciate your continued interest and thank you for your thoughtful question. We would like to clarify these issues in details.
> > >
> > > In REINFORCE, the issue of diminishing reward signals arises primarily due to the choice of **average baseline** in the advantage function $A(x, \tau) = r(x, \tau) - b(x)$, where $b(x)$ is chosen as the average reward of sampled solutions in most of the works because the ground truth is inaccessible in COPs, and using no baseline often leads to worse performance in practice. As the model approaches optimality, the reward differences shrink, leading to minimal advantage values.
> > >
> > > For example, consider the TSP-100 where the near-optimal policy samples 5 trajectories with lengths:
> > > [7.783,7.780,7.779,7.778,7.775], the advantage values will be [−0.004,−0.001,0,0.001,0.004], these small advantage values correspond to significant differences in the optimality gap but provide weak gradient signals for the policy.
> > > Consequently, REINFORCE struggles to sufficiently prioritize the best solutions.
> > >
> > > PO addresses this issue by utilizing qualitative information derived from pairwise comparisons between trajectories.
> > > Instead of relying on absolute reward differences, PO focuses on the **relative ranking** of solutions, which is invariant to the reward scale. PO maintains discriminable learning signals even when reward differences are minimal or maximal, enabling the policy to assign significantly higher probabilities to superior solutions.
> > >
> > > The distribution of advantage scale is illustrated in Figure 2, REINFORCE exhibits a narrow, peaked distribution around zero, indicating limited differentiation between trajectories. In contrast, PO displays a broader distribution, encompassing a wider range of positive and negative values. By emphasizing relative preferences, PO effectively directs the policy to favor better solutions.
> > >
> > > This alignment ensures that the training objective more closely matches the inference goal of selecting the best solutions, thereby mitigating the mismatch between training and inference objectives.
> > >
> > > **AQ2: Integration of local search may introduce off-policy issues?**
> > >
> > > **R:** Thank you for this insightful question. We would like to clarify the rationale behind using local search during the finetuning phase and address the potential off-policy concerns.
> > >
> > > **Why local search is used only for finetuning but not for whole training?**
> > >
> > > Firstly, we would like to clarify that the finetune stage is **optional**.
> > >
> > > - **Avoiding Large Distribution Shifts:** Applying LS during initial training would introduce significant changes to the solutions, causing a large distribution shift. This could destabilize training, as the policy would need to make drastic adjustments to align with modified solutions.
> > >
> > > - **Minor Adjustments:** After training, the policy already generates near-optimal solutions. Using 2-Opt only make minor refinements, modifying only a small segment (2–3%) of the solution on TSP100. This results in minimal distribution shift, allowing the policy to adjust smoothly.
> > >
> > > Empirically, we observe that integrating LS during finetuning improves performance without causing instability, suggesting that any off-policy bias does not significantly impact learning.
> > >
> > > **Why REINFORCE needs to address off-policy issues, but PO does not?**
> > >
> > > REINFORCE is an on-policy algorithm requiring samples from the current policy distribution. Using LS-modified solutions introduces off-policy data, necessitating importance sampling to correct for distribution mismatch under the **policy gradient framework**.
> > >
> > > In the finetuning phase, PO's learning objective naturally align with the **imitation learning framework** like BC [1] and DAgger [2].
> > > The loss function is: $ L_{finetune}(\theta) = f\left( \alpha \left[ \log \pi_{\theta}(\text{LS}(\tau) \mid x) - \log \pi_{\theta}(\tau \mid x) \right] \right) $. PO treats LS-modified solutions as expert demonstrations, allowing the policy to imitate them without off-policy issues.
> > >
> > > In summary, PO leverages the strengths of imitation learning during finetuning, allowing the policy to learn from LS-improved solutions without introducing significant off-policy bias. This approach ensures stable and effective learning, enhancing performance while maintaining theoretical soundness.
> > >
> > > References:
> > >
> > >     [1] Pomerleau, D. A. (1989). "ALVINN: An Autonomous Land Vehicle in a Neural Network." Advances in Neural Information Processing Systems.
> > >     [2] Ross, S., Gordon, G. J., & Bagnell, D. (2011). A reduction of imitation learning and structured prediction to no-regret online learning. In Proceedings of the fourteenth international conference on artificial intelligence and statistics (pp. 627-635).
> > >
> > > We hope this explanation clarifies the rationale behind our methodology and addresses your concerns. Please feel free to reach out if you have further questions.

---

> > > > ### Comment · Reviewer_trEc · 2024-11-27
> > > >
> > > > Thank you for answering my questions in detail. Your responses seem to have addressed my concerns. I’ll give you one more point. I hope for great results!

---

> > > > > ### Author Response · Authors · 2024-11-27
> > > > >
> > > > > Dear Reviewer trEc,
> > > > >
> > > > > Thank you for your thoughtful review and for taking the time to thoroughly consider our responses. We sincerely appreciate the higher score you assigned and are delighted that our answers addressed your concerns. Your feedback has been invaluable in refining our work, and your support means a great deal to us!
> > > > >
> > > > > Thank you once again for your encouragement.
> > > > >
> > > > > Best regards,
> > > > >
> > > > > The Authors.

---

### Author Response · Authors · 2024-11-22
**Summary of Revisions**

Dear Reviewers,

We sincerely appreciate the insightful reviews and constructive suggestions from all of you, which have greatly enhanced the quality of our submission. We are encouraged by the positive comments on our theoretical foundation and the practical effectiveness of our algorithm.

In this revised version, we have carefully addressed all feedback and incorporated the suggested changes into the main body, with additional content provided in the appendix due to page limitations. **The changes are highlighted in blue**, and the major modifications are summarized as follows:

**Summary of Changes:**

- **Additional experiments on different types of COPs**: We have included experiments on the Flexible Flow Shop Problem (FFSP) in Section 4.3 and Table 2.
- **Experiments on larger problem sizes**: Results for TSP instances with 500, 1,000, and 10,000 nodes are now included in Appendix F.2.
- **illustration of the PO framework**: Additional explanations and illustrations are provided in Appendix A..
- **Expanded discussion on convergence speed**: We have added a discussion on how PO accelerates convergence speed in Section 4.1.
- **Improved clarity and terminology**: The clarity of the claims and terms w.r.t _symbols_, _objective mismatch_,  _multi start mechanism_, and the _optional fine-tune phase_ are improved in the main body.

We hope that these revisions address your concerns and enhance the clarity and completeness of our paper.
Thank you again for your valuable feedback.
Please feel free to reach out if you have any further questions or require additional information.

Sincerely,

The Authors

---

### Note · Authors · 2025-01-23

I have read and agree with the venue's withdrawal policy on behalf of myself and my co-authors.